# FROB: Few-shot ROBust Model for Classification with Out-of-Distribution Detection

## Abstract

Nowadays, classification and Out-of-Distribution (OoD) detection in the few-shot setting remain challenging aims mainly due to rarity and the limited samples in the few-shot setting, and because of adversarial attacks. Accomplishing these aims is important for critical systems in safety, security, and defence. In parallel, OoD detection is challenging since deep neural network classifiers set high confidence to OoD samples away from the training data. To address such limitations, we propose the Few-shot ROBust (FROB) model for classification and few-shot OoD detection. We devise a methodology for improved robustness and reliable confidence prediction for few-shot OoD detection. We generate the support boundary of the normal class distribution and combine it with few-shot Outlier Exposure (OE). We propose a self-supervised learning few-shot confidence boundary methodology based on generative and discriminative models, including classification. The main contribution of FROB is the combination of the generated boundary in a self-supervised learning manner and the imposition of low confidence at this learned boundary. FROB implicitly generates strong adversarial samples on the boundary and forces samples from OoD, including our boundary, to be less confident by the classifier. FROB achieves generalization to unseen anomalies and adversarial attacks, with applicability to unknown, in the wild, test sets that do not correlate to the training datasets. To improve robustness, FROB redesigns and streamlines OE to work even for zero-shots. By including our learned boundary, FROB effectively reduces the threshold linked to the model's few-shot robustness, and maintains the OoD performance approximately constant and independent of the number of few-shot samples. The few-shot robustness analysis evaluation of FROB on different image sets and on One-Class Classification (OCC) data shows that FROB achieves competitive state-of-the-art performance and outperforms benchmarks in terms of robustness to the outlier OoD few-shot sample population and variability.

## 1 Introduction

In real-world settings, it is crucial to robustly perform classification and OoD detection with high levels of confidence. The problem of detecting whether a sample is in-distribution, from the training distribution, or OoD is critical for adversarial attacks. This is crucial nowadays in many applications in safety, security, and defence. However, deep neural networks produce overconfident predictions and do not distinguish in- and out-of-data-distribution. Adversarial examples, when small modifications of the input appear, can change the classifier decision. It is an important property of a classifier to address such limitations with high level of confidence, and provide robustness guarantees for neural networks. In parallel, OoD detection is a challenging aim since classifiers set high confidence to OoD samples away from the training data. The state-of-art models are overconfident in their predictions, and do not distinguish in- and OoD. The setting that our proposed Few-shot ROBust (FROB) model addresses is robust few-shot Out-of-Distribution (OoD) detection and few-shot Outlier Exposure (OE). To address rarity and the limited samples in the few-shot setting, we aim at reducing the number of the few-shots of the OoD samples, while maintaining accurate and robust performance.

Diverse data are available today in large quantities. Deep learning magnifies the difficulty of distinguishing OoD from in-distribution. It is possible to use such data to improve OoD detection by training detectors with auxiliary outlier sets (Hendrycks et al., 2019). OE enables detectors to generalize to detect unseen OoD samples with improved robustness and performance. Models trained

with different outliers can detect unmodelled data and improve OoD detection by learning cues for whether inputs are unmodelled. By exposing models to different OoD, the complement of the support of the normal class distribution is modelled and the detection of new types of anomalies is enabled. OE improves the calibration of deep neural network classifiers in the setting where a fraction of the data is OoD, addressing the problem of classifiers being overconfident when applied to OoD (Bitterwolf et al., 2020). Aiming at solving the few-shot robustness problem with classification and OoD detection, the contribution of our FROB methodology is the development of an integrated robust framework for self-supervised few-shot negative data augmentation on the distribution confidence boundary, combined with few-shot OE, for improved OoD detection. The combination of the generated boundary in a self-supervised learning way and the imposition of low confidence at this learned boundary is the main contribution of FROB, which greatly and decisively improves robustness for few-shot OoD detection. To address the rarity of relevant outliers during training using OoD samples, we propose to use even few-shots to improve the OoD detection performance. FROB achieves significantly better robustness and resilience to few-shot OoD detection, while maintaining competitive in-distribution accuracy. FROB achieves generalization to unseen anomalies, with applicability to new, in the wild, test sets that do not correlate to the training sets. FROB's evaluation on different sets, CIFAR-10, SVHN, CIFAR-100, and low-frequency noise, using cross-dataset and One-Class Classification (OCC) evaluations, shows that our self-supervised model with few-shot OE on the confidence boundary and few-shot adaptation improves the few-shot OoD detection performance and outperforms benchmarks. The robustness performance analysis of FROB to the number of few-shots and to outlier variation shows that it is robust to few-shots and outperforms baselines.

## 2 OUR PROPOSED FEW-SHOT ROBUSTNESS (FROB) METHODOLOGY

We propose FROB for few-shot OoD detection and classification using discriminative and generative models. We devise a methodology for improved robustness and reliable confidence prediction, to force low confidence close and away from the data. To improve robustness, FROB generates strong adversarial samples on the boundary close to the normal class. It finds the boundary of the normal class, and it combines the self-supervised learning few-shot boundary with our robustness loss.

**Flowchart of FROB.** Fig. 1 shows the flowchart of FROB which uses a discriminative model for classification and OoD detection. FROB also uses a generator for the OoD samples and the learned boundary. It generates low-confidence samples and performs active negative training with the generated OoD samples on the boundary. It performs self-supervised learning negative sampling of confidence boundary samples via the generation of strong and specifically adversarial OoD. It trains classifiers and generators to robustly classify as less confident samples on and out of the boundary.

**Our proposed loss.** We denote the normal class data by $\mathbf{x}$ where $\mathbf{x}_i$ are the labeled data with class labels $y_i$. Our proposed loss of the discriminative model which is minimized during training is

$$\arg\min_f \; -\frac{1}{N}\sum_{i=1}^{N}\log\frac{\exp(f_{y_i}(\mathbf{x}_i))}{\sum_{k=1}^{K}\exp(f_k(\mathbf{x}_i))} - \lambda\frac{1}{M}\sum_{m=1}^{M}\log\left(1-\frac{\exp(f(\mathbf{Z}_m))}{\sum_{k=1}^{K}\exp(f_k(\mathbf{Z}_m))}\right) \quad (1)$$

where $f(.)$ is the Convolutional Neural Network (CNN) discriminative model for multi-class classification with $K$ classes. Our loss has 2 terms and a hyper-parameter. The 2 losses operate on different samples for positive and negative training, respectively. The first loss is the cross-entropy between $y_i$ and the predictions, $\text{softmax}(f(\mathbf{x}_i))$; the CNN is followed by the normalized exponential to obtain the probability over the classes. Our robustness loss forces $f(.)$ to accurately detect outliers, in addition to classification. It operates on the few-shot OE samples, $\mathbf{Z}$. It is weighted by the hyper-parameter $\lambda$. $k$ is a class index. For the in-distribution, $N$ is the batch size and $i$ is the batch data sampling index. For the OoD, $M$ is the batch size and $m$ is the batch data sampling index.

FROB then trains a generator to generate low-confidence samples on the normal class boundary. Our algorithm includes these learned low-confidence samples in the training to improve the performance in the few-shot setting. Instead of using a large OE set, which constitutes an ad hoc choice of outliers to model the complement of the support of the normal class distribution, FROB performs learned negative data augmentation and self-supervised learning, to model the boundary of the support of the normal class distribution. We train a CNN deep neural network generator and denote it by $O(\mathbf{z})$, where $O$ refers to OoD samples and $\mathbf{z}$ are latent space samples from a standard Gaussian distribution.

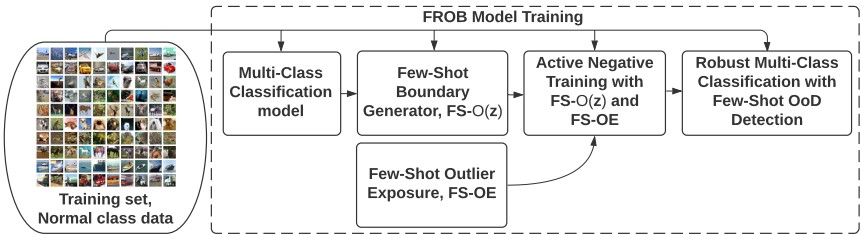

Figure 1: FROB training with learned negative sampling, FS-$O(\mathbf{z})$, and few-shot outliers, FS-OE.

Our proposed optimization of maximizing dispersion subject to being on the boundary is given by

$$\arg\ \min_O\ \frac{1}{N-1}\sum_{j=1,\ \mathbf{z}_j\neq\mathbf{z}}^{N}\frac{||\mathbf{z}-\mathbf{z}_j||_2}{||O(\mathbf{z})-O(\mathbf{z}_j)||_2}$$
$$+\ \mu\max_{l=1,2,...,K}\frac{\exp(f_l(O(\mathbf{z}))-f_l(\mathbf{x}))}{\sum_{k=1}^{K}\exp(f_k(O(\mathbf{z}))-f_k(\mathbf{x}))}\ +\ \nu\min_{j=1,2,...,Q}||O(\mathbf{z})-\mathbf{x}_j||_2 \qquad (2)$$

where using (2), we penalize the probability that $O(\mathbf{z})$ have higher confidence than the normal class. We hence make $O(\mathbf{z})$ have lower probability than $\mathbf{x}$ (Jolicoeur-Martineau, 2019; Ren et al., 2021).

FROB includes the learned low-confidence samples in the training by performing (1) with the self-generated few-shot boundary, $O(\mathbf{z})$, in addition to $\mathbf{Z}$. Our self-supervised learning mechanism to calibrate confidence in unforeseen scenarios is (2) followed by (1). FROB performs boundary data augmentation in a learnable self-supervised learning manner. It introduces self-generated boundary samples, and sets them as OoD to better perform few-shot OoD detection. This learned boundary has strong and adversarial anomalies close to the distribution support and near high probability normal class samples. FROB introduces optimal, relevant, and useful anomalies to more accurately detect few-shots of OoD (Wang et al., 2020a;b). It detects OoD robustly, by generating strong adversarial OoD samples and helpful task-specific anomalies. A property of our nested optimization, where the inner optimization is $O(\mathbf{z})$ in (2) and the outer one is cross-entropy with negative training in (1), is that if an optimum is reached for the inner one, an optimum will also be reached for the outer.

FROB addresses the few-shots problem by performing negative data augmentation in a well-sampled manner on the support boundary of the normal class. It performs OoD sample description and characterization, not allowing space between the normal class and our self-generated anomalies. FROB addresses the question of what OoD samples to introduce to our model for negative training, to robustly detect few-shots of data. FROB introduces self-supervised learning and learned data augmentation using the Deep Tightest-Possible Data Description algorithm of (2) followed by (1), and our self-generated confidence boundary in (2) is robust to mode collapse (Dionelis et al., 2020b;a). By performing scattering, FROB achieves diversity using the ratio of distances in the latent and data spaces rather than maximum entropy (von Kügelgen et al., 2021; Dieng et al., 2019). Our framework uses data space point-set distances (Dionelis et al., 2020b;a; Jalal et al., 2017; Jordan et al., 2019).

**Inference.** The Anomaly Score (AS) of FROB for *any* queried test sample, $\tilde{\mathbf{x}}$, during inference is

$$\text{AS}(f,\tilde{\mathbf{x}})\ =\ \max_{l=1,2,...,K}\frac{\exp(f_l(\tilde{\mathbf{x}}))}{\sum_{k=1}^{K}\exp(f_k(\tilde{\mathbf{x}}))} \qquad (3)$$

where if the AS is smaller than a threshold $\tau$, i.e. AS $< \tau$, $\tilde{\mathbf{x}}$ is OoD. Otherwise, $\tilde{\mathbf{x}}$ is in-distribution.

## 3 RELATED WORK ON CLASSIFICATION WITH OoD DETECTION

**Outlier Exposure.** The OE method trains detectors with outliers to improve the OoD performance to detect unseen anomalies (Hendrycks et al., 2019). Using auxiliary sets, disjoint from train and test data, models learn better representations for OoD detection. Confidence Enhancing Data Augmentation (CEDA), Adversarial Confidence Enhancing Training (ACET), and Guaranteed OoD Detection (GOOD) tackle the problem of classifiers being overconfident at OoD samples (Bitterwolf

et al., 2020; Hein et al., 2019). Their aim is to force low confidence in a $l_\infty$-norm ball around each OoD sample where the prediction confidence is $\max_{k=1,2,\ldots,K} p_k(\mathbf{x})$ for the output $K$-class softmax (Sensoy et al., 2018; Hariharan & Girshick, 2017; Jeong & Kim, 2020). CEDA employs point-wise robustness (Bastani et al., 2016; Rosenfeld et al., 2020). GOOD finds worst-case OoD detection guarantees. The models are trained on OE sets, using the 80 Million Tiny Images reduced by the normal class. Disjoint distributions are used for positive and negative training, but the OoD samples for OE are chosen in an ad hoc way. In contrast, FROB performs learned negative data augmentation on the boundary of the normal class to streamline and redesign few-shot OE (and zero-shot OE).

**Human prior.** GOOD defines the normal class, then filters it out from the 80 Million Tiny Images. This filtering-out process of normality from the OE set is human-dependent. This modified dataset is set as anomalies. Next, GOOD learns the normal class and sets low confidence to these OoD. This process is data-dependent, not automatic, and feature-dependent (Dionelis et al., 2021; Sohn et al., 2021). In contrast, FROB eliminates the need for feature extraction and human intervention which is the aim of Deep Learning, as these do not scale. This filtering-out process is not practical and cannot be used in real-world scenarios as anomalies are not confined in finite closed sets (Sensoy et al., 2020). FROB avoids feature-, application-, and dataset-dependent processes. Our self-supervised boundary data augmentation obviates memorization, scalability, and data diversity problems arising from memory replay and prioritized experience replay (Zaheer et al., 2020; Pourreza et al., 2021).

**Learned OoD samples.** The Confidence-Calibrated Classifier (CCC) uses a GAN to create samples out of, but close to the normal class (Lee et al., 2018a). FROB substantially differs from CCC, as CCC finds a threshold and not the boundary. CCC uses the OE set, $U(\mathbf{y})$, where the labels follow a Uniform distribution, to compute this threshold. This is limiting as the threshold depends on $U(\mathbf{y})$, which is an ad hoc choice of outliers. In contrast, FROB finds the confidence boundary and does not use $U(\mathbf{y})$ to find this boundary. FROB streamlines OE and few-shot outliers. Our boundary is not a function of $U(\mathbf{y})$, as $U(\mathbf{y})$ is not necessary (Sohn et al., 2021). For negative training, CCC defines a closeness metric (KL divergence), and then penalizes this metric (Zaheer et al., 2020; Asokan & Seelamantula, 2020; Dionelis et al., 2021). CCC suffers from mode collapse as it does not perform scattering for diversity. The models in Lee et al. (2018a); Vernekar et al. (2019a;b) and Wang et al. (2018) perform confidence-aware classification. Self-Supervised outlier Detection (SSD) creates OoD samples in the Mahalanobis metric (Sehwag et al., 2021). It is not a classifier, as it performs OoD detection with OE. FROB achieves fast inference with (3), in contrast to Tack et al. (2020) which is slow during inference (Goldberger et al., 2005). Tack et al. (2020) does not address issues arising from detecting with nearest neighbors while using a different composite loss for training.

## 4 EVALUATION AND RESULTS

We evaluate FROB trained on different sets, CIFAR-10, SVHN, CIFAR-100, 80 Million Tiny Images, Uniform noise, and low-frequency noise, and we report the Area Under the Receiver Operating Characteristic Curve (AUROC), the Adversarial AUROC (AAUROC), and the Guaranteed AUROC (GAUROC) which uses $l_\infty$-norm perturbations for the OoD (Bitterwolf et al., 2020; Croce & Hein, 2020). For the evaluation of FROB, we test different combinations of normal class sets, OE datasets, few-shot outliers (FS-OE), the generated boundary (FS-$O(\mathbf{z})$), and test sets, in an alternating manner. We examine the generalization performance of FROB to few-shots of unseen new OoD samples at the dataset level, Out-of-Dataset anomalies. To examine the robustness to the number of few-shot samples, we decrease the number of few-shots by dividing them by two. We perform uniform sampling for choosing the few-shots, and we examine the variation of the dependent variable, AUROC, to changes of the independent variable, the provided number of few-shots of OoD (FS-OE). In this way, we evaluate the robustness of FROB to the number of few-shots. We also examine the Failure Point of our proposed FROB algorithm and of benchmarks; we define this Break Point as the number of few-shots from which the performance in AUROC decreases and then eventually falls to $0.5$.

**Datasets.** For normal class, we use CIFAR-10 and SVHN. For OE, we use 80 Million Tiny Images, SVHN, and CIFAR-100. For few-shot, we use CIFAR-10, CIFAR-100, SVHN, and Low-Frequency Noise (LFN). We evaluate FROB on CIFAR-100, SVHN, CIFAR-10, LFN, and Uniform noise.

**Benchmarks.** We compare FROB to benchmarks. Having access to large OE sets is not representative of the few-shot OoD detection setting. We compare FROB to GOOD, CEDA, ACET, and OE (Bitterwolf et al., 2020; Hein et al., 2019; Hendrycks et al., 2019). We also compare FROB to

GEOM, GOAD, DROCC, Hierarchical Transformation-Discriminating Generator (HTD), Support Vector Data Description (SVDD), and Patch SVDD (PaSVDD) in the few-shot setting using One-Class Classification (OCC) (Sheynin et al., 2021). GOOD and Hendrycks et al. (2019) use the 80 Million Tiny Images for OE. FROB outperforms baselines in the few-shot OoD detection setting.

**Ablation Study.** We test FROB for: (i) with OE, and without (w/o) FS-OE and FS-$O(\mathbf{z})$, (ii) with (w/) FS-OE and w/o FS-$O(\mathbf{z})$, (iii) w/ FS-OE and FS-$O(\mathbf{z})$, and (iv) w/ FS-OE, FS-$O(\mathbf{z})$, and OE.

## 4.1 FROB PERFORMANCE ANALYSIS COMPARED TO BENCHMARKS

**Overview.** We evaluate the benchmarks using OE and compare them to FROB and its OoD performance. We analyse the performance of baselines, CEDA, OE model, ACET, and GOOD, using 80 Million Tiny Images for OE, as well as the performance of CCC using OE SVHN or CIFAR-10. We compare them to FROB, using the 80 Million Tiny Images for OE. FROB without using our self-supervised generated distribution boundary shows similar behavior to the benchmarks, and outperforms them in all the examined AUC-type metrics in Table 1. Table 1 shows the results of FROB without few-shot boundary samples, when the normal class is SVHN and CIFAR-10, using the OE set 80 Million Tiny Images, evaluated on different test sets in AUROC. FROB without $O(\mathbf{z})$ outperforms benchmarks. Taking into account this FROB model behavior, we examine the performance of FROB without the boundary $O(\mathbf{z})$ to a variable number of few-shot outliers in Figs. 1 and 2. Without the generated $O(\mathbf{z})$, the AUROC performance decreases as the number of few-shots decreases, and is not robust and suitable for few-shot OoD detection, for few-shots less than approximately 800. Then, we examine the performance of FROB with the self-learned $O(\mathbf{z})$, in Figs. 4 and 5. Sec. 4.2 shows that $O(\mathbf{z})$ is effective and that FROB is robust, even to a very small number of few-shots.

### 4.1.1 FROB PERFORMANCE ANALYSIS USING OE

First, we examine the performance of benchmarks, CEDA, OE, ACET, and GOOD, when setting (a) SVHN and (b) CIFAR-10 as the normal class, with OE 80 Million Tiny Images, tested on different sets, CIFAR-100, CIFAR-10, SVHN, and Uniform noise, in AUROC, AAUROC, and GAUROC. We examine the performance of CCC using CIFAR-10 for OE, tested on Uniform noise. We compare the performance of benchmarks to that of FROB without $O(\mathbf{z})$. Our algorithm, when setting SVHN as the normal class, on average outperforms the baselines CCC, CEDA, OE, and ACET, in AAUROC and GAUROC, and yields competitive results in AUROC. FROB shows comparable performance to GOOD in Table 1. This is also shown in Table 3, in the Appendix. When setting CIFAR-10 as the normal class, FROB on average outperforms the benchmarks CEDA, OE, ACET, and GOOD in AUC-type metrics, according to Table 1 and also to Table 4 in the Appendix. Table 1, as well as Tables 3 and 4 in the Appendix, present the performance of FROB without using our self-generated boundary samples, $O(\mathbf{z})$. They, together with Table 6, show that FROB outperforms benchmarks.

### 4.1.2 COMPARISON OF FROB WITH BENCHMARKS

**Analysis of FROB without $O(\mathbf{z})$ by reducing the few-shot outliers.** Fig. 2 shows the performance of FROBInit for normal CIFAR-10 without $O(\mathbf{z})$, using variable number of Outlier Set SVHN few-shots, tested on different sets. The FROBInit performance decreases with reducing number of SVHN few-shots. A low AUROC of $0.5$ is reached for approximately $800$ few-shots for CIFAR-100, in Table 5 in the Appendix. Fig. 2 shows that the benchmarks in Table 1, as well as in Tables 3 and 4 in the Appendix, do not achieve a robust performance for decreasing few-shots, their performance is reduced with decreasing number of few-shots, yield a steep decline for few-shots less than $1830$ samples, and have a Failure Break Point at approximately $800$ few-shots for the test set CIFAR-100 (AUROC $0.51$). The performance of FROBInit, without $O(\mathbf{z})$, decreases fast and relatively sharp below $1830$ samples, when testing on low-frequency noise in Fig. 2, where for few-shots less than $1800$, the modeling error covering the full complement of the support of the normal class is high.

Fig. 3 and Table 5 show the performance of FROBInit in GAUROC for normal CIFAR-10 without $O(\mathbf{z})$, with variable number of FS SVHN. The GAUROC performance of FROB shows that when the few-shots originate from the test set, SVHN, a less steep decrease is observed compared to when the few-shots and the test samples are from different sets, low-frequency noise. The rate of decrease of GAUROC below $1800$ samples is small when the OE and the test samples are from the same set. The GAUROC performance of FROB without $O(\mathbf{z})$, for the test set CIFAR-100, is

Table 1: Performance of benchmarks with 80 Million Tiny Images in AUROC, AAUROC, and GAUROC (Bitterwolf et al., 2020). Comparison to FROBInit without $O(\mathbf{z})$. *FROBInit refers to FROB w/o $O(\mathbf{z})$, C10 to CIFAR-10, C100 to CIFAR-100, 80M to 80 Million Tiny Images, and UN to Uniform noise.*

| NORMAL | MODEL | OUTLIER | TEST DATA | AUROC | AAUROC | GAUROC |
|---|---|---|---|---|---|---|
| SVHN | FROBINIT | 80M | C10, C100, UN | 0.995 | 0.995 | 0.979 |
| SVHN | CCC | C10 | C10, UN | **1.000** | 0.000 | 0.000 |
| SVHN | CEDA | 80M | C10, C100, UN | 0.999 | 0.773 | 0.000 |
| SVHN | OE | 80M | C10, C100, UN | **1.000** | 0.736 | 0.000 |
| SVHN | ACET | 80M | C10, C100, UN | 0.999 | 0.984 | 0.000 |
| SVHN | GOOD | 80M | C10, C100, UN | 0.998 | **0.987** | **0.984** |
| C10 | FROBINIT | 80M | SVHN, C100, UN | 0.860 | 0.860 | **0.718** |
| C10 | CCC | SVHN | SVHN, UN | 0.570 | 0.000 | 0.000 |
| C10 | CEDA | 80M | SVHN, C100, UN | 0.957 | 0.427 | 0.000 |
| C10 | OE | 80M | SVHN, C100, UN | **0.962** | 0.522 | 0.000 |
| C10 | ACET | 80M | SVHN, C100, UN | 0.957 | 0.871 | 0.000 |
| C10 | GOOD | 80M | SVHN, C100, UN | 0.817 | 0.709 | 0.700 |

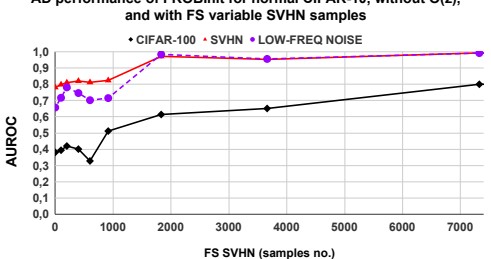 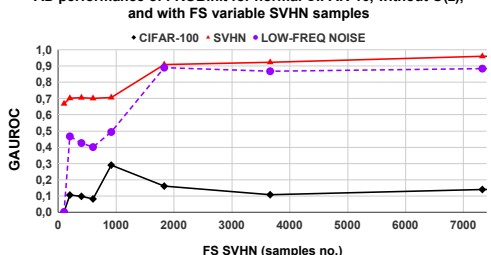

Figure 2: OoD performance of FROBInit in AUROC, for normal class CIFAR-10, w/o $O(\mathbf{z})$ and w/ few-shots of variable number from SVHN.

Figure 3: Performance of FROBInit in GAUROC for normal CIFAR-10 w/o $O(\mathbf{z})$ and w/ variable number of FS samples from SVHN.

low. FROBInit achieves better performance than the benchmarks in the FS OoD detection setting (Table 1). FROBInit with only few-shots, the OoD performance rapidly decreases for FS of 1800 samples, and tends to a Break Point of approximately 800 shots for normal CIFAR-10. It shows a more robust behavior at reducing the number of FS when the Outlier Set and test sets are the same.

**Using reduced number of 80 Million.** Table 6 in the Appendix shows the performance of FROBInit using the 80 Million Tiny Images for Outlier Dataset with all available data and for reduced number of samples, few-shots of it (FS). We evaluate FROBInit without the boundary, $O(\mathbf{z})$, on different test sets, CIFAR-10, CIFAR-100, SVHN, low-frequency noise, uniform noise, and 80 Million Tiny Images. We set SVHN and CIFAR-10 as the normal classes. The performance of FROBInit slightly decreases, in AUC-type metrics, when 73257 samples from 80 Million Tiny Images are used as the chosen Outlier Calibration Set, instead of the 50 million total training samples, during training.

**Using various Outlier Sets.** Table 7, in the Appendix, shows the performance of FROBInit trained on normal CIFAR-10 with the Outlier Datasets -all data- of SVHN, CIFAR-100, and 80 Million Tiny Images 73257 samples, without our $O(\mathbf{z})$, tested on different sets in AUC-type metrics. FROB, using SVHN for Outlier Dataset, achieves higher AUROC compared to using the OE sets 80 Million Tiny Images and CIFAR-100. FROBInit achieves higher GAUROC with the Outlier Dataset set SVHN, i.e. average 0.69, and 80 Million Tiny Images, average 0.66, compared to when using CIFAR-100.

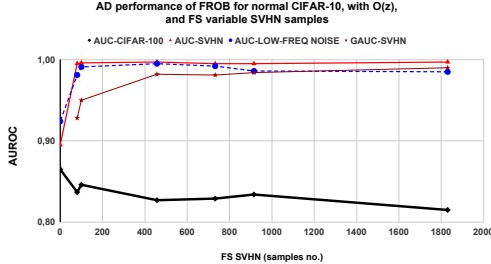 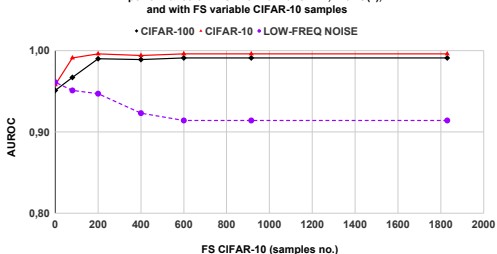

Figure 4: FROB for normal C10 with $O(\mathbf{z})$ and FS SVHN, in AUROC and GAUROC.

Figure 5: FROB performance in AUROC for normal class SVHN w/ $O(\mathbf{z})$ and FS C10.

Table 2: Mean OCC performance of FROB w/ $O(\mathbf{z})$, w/ 80 FS OCC C10 (Sheynin et al., 2021).

| NORMAL | MODEL | $O(\mathbf{z})$ | TEST DATA | AUROC |
|---|---|---|---|---|
| C10: OCC | FROB | w/ | C10: OCC | **0.784** |
| C10: OCC | FROB W/ OUTLIER SVHN | w/ | C10: OCC | **0.802** |
| C10: OCC | HTD (SHEYNIN ET AL., 2021) | w/o | C10: OCC | 0.756 |
| C10: OCC | GEOM (WHICH > GOAD) | w/o | C10: OCC | 0.735 |
| C10: OCC | SVDD (> PASVDD, DROCC) | w/o | C10: OCC | 0.608 |

## 4.2 EFFECTIVENESS OF OUR PROPOSED CONFIDENCE BOUNDARY

**Efficacy and effectiveness of learned boundary.** We evaluate FROB with the generated boundary, $O(\mathbf{z})$, and few-shot outliers. Fig. 4, and Table 8 in the Appendix, show the performance of FROB trained on normal class CIFAR-10 with $O(\mathbf{z})$ and few-shots from SVHN in decreasing number. We evaluate the performance of FROB on the test sets CIFAR-100, SVHN, and low-frequency noise. Compared to Fig. 2, when we use $O(\mathbf{z})$, the performance increases, showing robustness even for a very small number of few-shots, even for zero-shots. We have experimentally demonstrated the effectiveness of $O(\mathbf{z})$, in Fig. 4. We have demonstrated the improvement in AUROC, when $O(\mathbf{z})$ is used, compared to when it is not used. FROB using few-shot data outperforms benchmarks and improves robustness to the number of few-shots, pushing down the phase transition point (Fig. 4).

**FROB robustness to number of few-shots.** With decreasing few-shot samples, the performance of FROB in AUROC is robust and approximately independent of the FS number of samples, till approximately zero-shots. FROB achieves robustness using the boundary $O(\mathbf{z})$ as the performance is approximately independent of the FS samples for the test sets, CIFAR-100, SVHN, and low-frequency noise. The component of FROB with the highest benefit is $O(\mathbf{z})$. FROB achieves robustness to the number of few-shot samples, and this is the main contribution of this paper. As the number of few-shots decreases, the performance of FROB does not decrease. When the few-shots are from the test set, Fig. 4 (and Table 8 in Appendix) shows that using the learned boundary, $O(\mathbf{z})$, is effective and robust in the few-shot OoD detection setting. FROB with the self-generated boundary samples, $O(\mathbf{z})$, achieves better performance than the benchmarks in the few-shot OoD detection setting.

**Existing methodologies sensitive to number of few-shots.** Current methodologies are not robust to a small number of few-shots as they perform negative training by including OoD samples randomly somewhere in the data space, allowing a lot of unfilled space between the OoD and the normal class (Bitterwolf et al., 2020; Hendrycks et al., 2019). They need 50 million outliers, to model the complement of the support of the normal class distribution. They use irrelevant conservative OoD samples and do not model the support boundary of the normal class distribution. Instead, FROB learns OoD samples generated on the boundary, not requiring 50 million outliers. FROB redesigns and streamlines OE, to work even for zero-shots. Our negative data augmentation creates the tightest possible OoD samples that are as close as possible to the support of the normal class distribution.

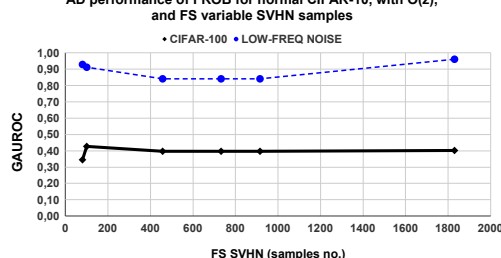 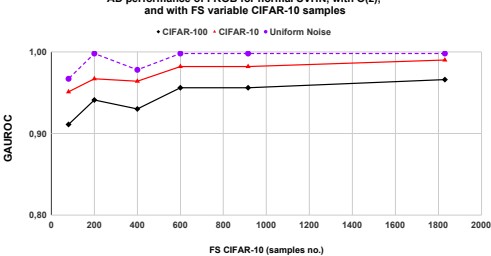

Figure 6: FROB performance in GAUROC for normal CIFAR-10 w/ $O(\mathbf{z})$ and variable number of OE SVHN, tested on the unseen sets CIFAR-100 and low frequency noise, using OE 80M.

Figure 7: OoD performance of FROB for normal SVHN in GAUROC using the boundary, $O(\mathbf{z})$, and variable number of FS-OE CIFAR-10, tested on CIFAR-100, CIFAR-10, and Uniform noise.

### 4.2.1 OoD Detection Performance of FROB when FS data are from Test Set

FROB improves the AUROC and the GAUROC, when the few-shots and the OoD test samples originate from the same set. In addition to improving both the AUROC and the GAUROC when the few-shot outliers originate from the test set, as shown in Figs. 4 and 2, our proposed FROB model also improves the AUROC performance, when the few-shots and the OoD test samples originate from different sets. These results for FROB are also presented in Tables 8 and 5 in the Appendix.

Tables 10 and 11 in the Appendix show that FROB (i) improves both the AUROC and the GAUROC when the few-shots and the OoD test samples originate from the same set, and (ii) enhances the AUROC when the few-shots and the OoD test samples originate from different sets, i.e. CIFAR-100 and low frequency noise. When using the learned boundary $O(\mathbf{z})$ and FS, as well as the 80 Million Tiny Images, the performance of FROB in GAUROC improves, according to Tables 7-10 in the Appendix for $O(\mathbf{z})$ of 1830, 915, 100, and 80 samples, respectively. When using the same Outlier Dataset and test set, SVHN, without the 80 Million Tiny Images set, FROB achieves a comparable high value in GAUROC, 0.98 in Fig. 4, compared to when using 80 Million Tiny Images, 0.97.

**FROB outperforming baselines.** FROB with $O(\mathbf{z})$ achieves an AUROC of 0.92 for normal CIFAR-10, SVHN 1830 samples, and test set Uniform noise in Table 8 in the Appendix. It is effective and outperforms benchmarks. FROB outperforms Lee et al. (2018a): for normal CIFAR-10, when the Outlier Dataset is SVHN, CCC yields an AUROC of 0.14 for Uniform noise. CCC does not have few-shot capability, FS functionality. It does not test CIFAR-100 (small domain gap to CIFAR-10).

Fig. 4 shows the performance of FROB in GAUROC for normal CIFAR-10 with $O(\mathbf{z})$, using a variable number of FS data SVHN, tested on SVHN. We evaluate FROB with $O(\mathbf{z})$, by reducing the number of few-shot outliers. The GAUROC performance shows that when the few-shots originate from the test set, SVHN, a less steep decrease is achieved compared to when the few-shots and the test samples are from different sets, i.e. SVHN and low-frequency noise. The rate of decrease of the GAUROC of FROB below 1830 samples is smaller when the few-shots are from the test dataset.

Figure 5, together with Table 14 and Figure 7 in the Appendix, show the performance of FROB using $O(\mathbf{z})$, the normal class SVHN, and variable number of FS CIFAR-10 samples. In Figures 5 and 7, compared to Figure 4, we show that FROB achieves better performance for normal class SVHN, compared to for normal CIFAR-10 in all AUC-type metrics on the unseen test dataset CIFAR-100.

### 4.3 Performance of FROB on Useen, in the Wild, Datasets

We evaluate FROB using OoD test samples from unseen, in the wild, sets. Importantly, we evaluate FROB on test samples that are neither from the normal class nor from the few-shot data (or from Outlier Dataset). In Figs. 2 and 4, we show the performance of FROB in the few-shot setting, for the normal CIFAR-10 with few-shot outliers from SVHN, tested on the unknown sets of CIFAR-10 and low-frequency noise. Fig. 4 shows that the OoD detection performance of FROB in the few-shot setting is robust, tested on the new sets of CIFAR-100 and low frequency noise. FROB has a robust performance in AUROC with reduced number of SVHN few-shots, till approximately zero-shots.

According to Table 8 in the Appendix, the GAUROC performance of FROB depends on the unknown test set and obtains low values for SVHN 80 few-shots, approximately 0.07 on CIFAR-100, 0.25 on low frequency noise, and 0.02 on Uniform noise. We use an effective Outlier Set, 80 Million Tiny Images, to improve the GAUROC of FROB in the few-shot setting to obtain Table 9: for SVHN 80 few-shots, 0.43 on CIFAR-100, 0.95 on low frequency noise, and 0.79 on Uniform noise.

**Effect of domain, normal class.** The performance of FROB with the boundary in AUROC depends on the normal class. In Table 14 and Fig. 5, the performance of FROB with $O(\mathbf{z})$ is higher for normal SVHN than for CIFAR-10, as presented in Fig. 4 and Table 8. For zero-shots, using the CIFAR-100 test set, the AUROC is 0.87 when the normal class is CIFAR-10 compared to 0.95 when the normal class is SVHN. Figs. 5 and 7 show that FROB is robust and effective for normal SVHN, for Outlier Set CIFAR-10, on seen and unseen sets. FROB with $O(\mathbf{z})$ is not sensitive to the number of few-shots in Figs. 4-7 in the FS OoD detection setting when we have OoD sample complexity constraints.

### 4.3.1 FROB PERFORMANCE WITH $O(\mathbf{z})$ IN GAUROC USING 80 MILLION TINY IMAGES

To improve the performance of FROB with the learned boundary in GAUROC, we use the 80 Million Tiny Images for OE. Fig. 6 and Table 9 both show that the performance of FROB, in GAUROC, improves for normal CIFAR-10 with $O(\mathbf{z})$ and variable number of OE SVHN, tested on the unseen sets CIFAR-100 and low frequency noise. In the few-shot OoD detection setting, for decreasing few-shots till approximately 80, FROB achieves a mean value of approximately 0.88 in GAUROC, tested on low frequency noise. FROB with the OE 80 Million Tiny Images set and the learned $O(\mathbf{z})$ reduces the threshold linked to the model's few-shot robustness. Tables 10-13 in the Appendix show the performance of FROB with $O(\mathbf{z})$, in GAUROC (and AUROC), when the OE 80 Million Tiny Images in reduced number (73257) is used and few-shots from 1830 to 80 are used. The performance of FROB in GAUROC increases with OE 80 Million Tiny Images, compared to without this set.

### 4.4 EVALUATION OF FROB USING OCC COMPARED TO BENCHMARKS

We evaluate FROB using OCC for each CIFAR-10 class. We report the mean performance of FROB and compare FROB to benchmarks, HTD, GEOM, GOAD, DROCC, SVDD, and PaSVDD, in the few-shot OE setting of 80 samples (Sheynin et al., 2021). FROB outperforms benchmarks in Table 2, and in Table 15 in the Appendix. Tables 2 and 15 show the performance of FROB in the OCC setting, when normality is a CIFAR-10 class. FROB using the self-learned boundary outperforms baselines in the few-shot OoD detection task, when we have budget constraints and OoD sampling complexity limitations. The improvement of our proposed FROB model, using OE SVHN, in mean AUROC is 2.3%, when compared to without using OE SVHN, in Table 2, and in Table 15 in the Appendix.

## 5 CONCLUSION

We have proposed FROB which uses the generated support boundary of the normal data distribution for few-shot OoD detection. FROB tackles the few-shot problem using classification with OoD detection. The contribution of FROB is the combination of the generated boundary in a self-supervised learning manner and the imposition of low confidence at this learned boundary. To improve robustness, FROB generates strong adversarial samples on the boundary, and forces samples from OoD and on the boundary to be less confident. By including the self-produced boundary, FROB reduces the threshold linked to the model's few-shot robustness. FROB redesigns, restructures, and streamlines OE to work even for zero-shots. It robustly performs classification and few-shot OoD detection with a high level of reliability in real-world applications, in the wild. FROB maintains the OoD performance approximately constant, independent of the few-shot number. The performance of FROB with the self-supervised learning boundary is robust and effective, as the performance is approximately stable as the few-shot outliers decrease in number, while the performance of FROB without $O(\mathbf{z})$ decreases as the few-shots decrease. The evaluation of FROB, on many sets, shows that it is effective, achieves competitive state-of-the-art performance, and outperforms benchmarks in the few-shot OoD detection setting in AUC-type metrics. In the future, in addition to confidence and the class, FROB will also output important regions and bounding boxes around abnormal objects.

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

# APPENDIX

## EVALUATION TABLES

The main Tables used for the evaluation of our proposed FROB model are:

Table 3: OoD performance of benchmarks w/ Outlier Dataset 80 Million Tiny Images in AUROC, AAUROC, and GAUROC. Comparison to FROB w/o the self-supervised learning boundary, $O(\mathbf{z})$. *C10 refers to CIFAR-10, C100 to CIFAR-100, 80M to 80 Million Tiny Images, and UN to Uniform Noise.*

| NORMAL | MODEL | $O(\mathbf{z})$ | OUTLIER | TEST DATA | AUROC | AAUROC | GAUROC |
|--------|-------|--------|---------|-----------|-------|--------|--------|
| SVHN | CCC | w/o | C10 | C10 | 0.999 | 0.000 | 0.000 |
| SVHN | CCC | w/o | C10 | UN | 1.000 | 0.000 | 0.000 |
| SVHN | CEDA | w/o | 80M | C100 | 0.999 | 0.639 | 0.000 |
| SVHN | CEDA | w/o | 80M | C10 | 0.999 | 0.687 | 0.000 |
| SVHN | CEDA | w/o | 80M | UN | 0.999 | 0.993 | 0.000 |
| MEAN | CEDA | w/o | 80M | | 0.999 | 0.773 | 0.000 |
| SVHN | OE | w/o | 80M | C100 | 1.000 | 0.602 | 0.000 |
| SVHN | OE | w/o | 80M | C10 | 1.000 | 0.625 | 0.000 |
| SVHN | OE | w/o | 80M | UN | 1.000 | 0.982 | 0.000 |
| MEAN | OE | w/o | 80M | | 1.000 | 0.736 | 0.000 |
| SVHN | ACET | w/o | 80M | C100 | 1.000 | 0.994 | 0.000 |
| SVHN | ACET | w/o | 80M | C10 | 1.000 | 0.995 | 0.000 |
| SVHN | ACET | w/o | 80M | UN | 0.999 | 0.963 | 0.000 |
| MEAN | ACET | w/o | 80M | | 0.999 | 0.984 | 0.000 |
| SVHN | GOOD | w/o | 80M | C100 | 0.996 | 0.977 | 0.973 |
| SVHN | GOOD | w/o | 80M | C10 | 0.997 | 0.984 | 0.981 |
| SVHN | GOOD | w/o | 80M | UN | 1.00 | 0.999 | 0.998 |
| MEAN | GOOD | w/o | 80M | | 0.998 | 0.987 | 0.984 |
| SVHN | FROBINIT | w/o | 80M | C100 | 0.993 | 0.993 | 0.975 |
| SVHN | FROBINIT | w/o | 80M | C10 | 0.995 | 0.995 | 0.979 |
| SVHN | FROBINIT | w/o | 80M | UN | 0.997 | 0.997 | 0.984 |
| MEAN | FROBINIT | w/o | 80M | | **0.995** | **0.995** | **0.979** |

Table 4: OoD performance of benchmarks with 80 Million Tiny Images for Outlier Set, evaluated on different sets in AUROC, AAUROC, and GAUROC. Comparison with FROBInit without (w/o) the boundary samples, $O(\mathbf{z})$, as well as with FROB with (w/) $O(\mathbf{z})$ boundary. *Here, in this Table, C10 refers to CIFAR-10, C100 to CIFAR-100, 80M to 80 Million Tiny Images, and UN to Uniform Noise.*

| NORMAL | MODEL | $O(\mathbf{z})$ | OUTLIER | TEST | AUROC | AAUROC | GAUROC |
|--------|-------|-----------------|---------|------|-------|--------|--------|
| C10 | CCC | w/o | SVHN | SVHN | 0.999 | 0.000 | 0.000 |
| C10 | CCC | w/o | SVHN | UN | 0.141 | 0.000 | 0.000 |
| C10 | CEDA | w/o | 80M | C100 | 0.918 | 0.319 | 0.000 |
| C10 | CEDA | w/o | 80M | SVHN | 0.979 | 0.257 | 0.000 |
| C10 | CEDA | w/o | 80M | UN | 0.973 | 0.705 | 0.00 |
| MEAN | CEDA | w/o | 80M | | 0.957 | 0.427 | 0.000 |
| C10 | OE | w/o | 80M | C100 | 0.924 | 0.110 | 0.000 |
| C10 | OE | w/o | 80M | SVHN | 0.976 | 0.70 | 0.000 |
| C10 | OE | w/o | 80M | UN | 0.987 | 0.757 | 0.000 |
| MEAN | OE | w/o | 80M | | 0.962 | 0.522 | 0.000 |
| C10 | ACET | w/o | 80M | C100 | 0.907 | 0.745 | 0.000 |
| C10 | ACET | w/o | 80M | SVHN | 0.966 | 0.880 | 0.000 |
| C10 | ACET | w/o | 80M | UN | 0.997 | 0.989 | 0.000 |
| MEAN | ACET | w/o | 80M | | 0.957 | 0.871 | 0.000 |
| C10 | GOOD | w/o | 80M | C100 | 0.700 | 0.547 | 0.542 |
| C10 | GOOD | w/o | 80M | SVHN | 0.757 | 0.589 | 0.569 |
| C10 | GOOD | w/o | 80M | UN | 0.995 | 0.992 | 0.990 |
| MEAN | GOOD | w/o | 80M | | 0.817 | 0.709 | 0.700 |
| C10 | FROBINIT | w/o | 80M | C100 | 0.760 | 0.760 | 0.560 |
| C10 | FROBINIT | w/o | 80M | SVHN | 0.836 | 0.836 | 0.648 |
| C10 | FROBINIT | w/o | 80M | UN | 0.985 | 0.985 | 0.945 |
| MEAN | FROBINIT | w/o | 80M | | 0.860 | 0.860 | 0.718 |

Table 5: Robustness sensitivity analysis of FROBInit to the number of Outlier Datasets and FS samples. OoD detection performance of FROBInit using FS outliers of variable number and an Outlier Set, evaluated on various sets in AUROC, AAUROC, and GAUROC. *C10 refers to CIFAR-10, C100 to CIFAR-100, 80M to 80 Million Tiny Images, UN to Uniform Noise, and LFN to Low Frequency Noise.*

| NORMAL | MODEL | $O(\mathbf{z})$ | OUTLIER, FS | TEST | AUROC | AAUROC | GAUROC |
|--------|-------|------|-------------|------|-------|--------|--------|
| C10 | FROBINIT | W/O | SVHN:73257 | C100 | 0.829 | 0.829 | 0.153 |
| C10 | FROBINIT | W/O | SVHN: 3660 | C100 | 0.651 | 0.651 | 0.108 |
| C10 | FROBINIT | W/O | SVHN: 1830 | C100 | 0.614 | 0.614 | 0.161 |
| C10 | FROBINIT | W/O | SVHN: 915 | C100 | 0.512 | 0.512 | 0.290 |
| C10 | FROBINIT | W/O | SVHN: 600 | C100 | 0.328 | 0.328 | 0.082 |
| C10 | FROBINIT | W/O | SVHN: 400 | C100 | 0.401 | 0.401 | 0.098 |
| C10 | FROBINIT | W/O | SVHN: 200 | C100 | 0.420 | 0.420 | 0.106 |
| C10 | FROBINIT | W/O | SVHN: 100 | C100 | 0.394 | 0.394 | 0.002 |
| C10 | FROBINIT | W/O | SVHN: 0 | C100 | 0.381 | 0.381 | 0.000 |
| C10 | FROBINIT | W/O | SVHN:73257 | SVHN | 0.998 | 0.998 | 0.990 |
| C10 | FROBINIT | W/O | SVHN: 7325 | SVHN | 0.993 | 0.993 | 0.960 |
| C10 | FROBINIT | W/O | SVHN: 3660 | SVHN | 0.953 | 0.953 | 0.923 |
| C10 | FROBINIT | W/O | SVHN: 1830 | SVHN | 0.972 | 0.972 | 0.909 |
| C10 | FROBINIT | W/O | SVHN: 915 | SVHN | 0.824 | 0.824 | 0.706 |
| C10 | FROBINIT | W/O | SVHN: 600 | SVHN | 0.812 | 0.812 | 0.701 |
| C10 | FROBINIT | W/O | SVHN: 400 | SVHN | 0.820 | 0.820 | 0.705 |
| C10 | FROBINIT | W/O | SVHN: 200 | SVHN | 0.810 | 0.810 | 0.702 |
| C10 | FROBINIT | W/O | SVHN: 100 | SVHN | 0.798 | 0.798 | 0.668 |
| C10 | FROBINIT | W/O | SVHN: 0 | SVHN | 0.781 | 0.781 | 0.001 |
| C10 | FROBINIT | W/O | SVHN:73257 | LFN | 0.991 | 0.991 | 0.934 |
| C10 | FROBINIT | W/O | SVHN: 7325 | LFN | 0.992 | 0.992 | 0.884 |
| C10 | FROBINIT | W/O | SVHN: 3660 | LFN | 0.956 | 0.956 | 0.868 |
| C10 | FROBINIT | W/O | SVHN: 1830 | LFN | 0.984 | 0.984 | 0.890 |
| C10 | FROBINIT | W/O | SVHN: 915 | LFN | 0.715 | 0.715 | 0.494 |
| C10 | FROBINIT | W/O | SVHN: 600 | LFN | 0.702 | 0.702 | 0.401 |
| C10 | FROBINIT | W/O | SVHN: 400 | LFN | 0.746 | 0.746 | 0.426 |
| C10 | FROBINIT | W/O | SVHN: 200 | LFN | 0.781 | 0.781 | 0.467 |
| C10 | FROBINIT | W/O | SVHN: 100 | LFN | 0.717 | 0.717 | 0.002 |
| C10 | FROBINIT | W/O | SVHN: 0 | LFN | 0.657 | 0.657 | 0.000 |
| C10 | FROBINIT | W/O | SVHN: 915 | C100 | 0.512 | 0.512 | 0.209 |
| C10 | FROBINIT | W/O | SVHN: 915 | SVHN | **0.824** | **0.824** | **0.706** |
| C10 | FROBINIT | W/O | SVHN: 915 | LFN | 0.715 | 0.715 | 0.494 |
| C10 | FROBINIT | W/O | SVHN: 915 | UN | 0.811 | 0.811 | 0.555 |

Table 6: OoD performance of FROBInit w/o $O(\mathbf{z})$ and w/ the Outlier Set 80 Million Tiny Images.

| NORMAL | MODEL | $O(\mathbf{z})$ | OUTLIER | TEST | AUROC | AAUROC | GAUROC |
|--------|-------|-------|---------|------|-------|--------|--------|
| SVHN | FROBINIT | W/O | 80M | C100 | 0.993 | 0.993 | 0.975 |
| SVHN | FROBINIT | W/O | 80M | C10 | 0.995 | 0.995 | 0.979 |
| SVHN | FROBINIT | W/O | 80M | LFN | 0.963 | 0.963 | 0.791 |
| SVHN | FROBINIT | W/O | 80M | UN | 0.997 | 0.997 | 0.984 |
| SVHN | FROBINIT | W/O | 80M | 80M | 0.994 | 0.994 | 0.979 |
| MEAN | FROBINIT | W/O | 80M | | **0.988** | **0.988** | **0.942** |
| SVHN | FROBINIT | W/O | 80M: 73257 | C100 | 0.992 | 0.992 | 0.976 |
| SVHN | FROBINIT | W/O | 80M: 73257 | C10 | 0.994 | 0.994 | 0.982 |
| SVHN | FROBINIT | W/O | 80M: 73257 | LFN | 0.958 | 0.958 | 0.763 |
| SVHN | FROBINIT | W/O | 80M: 73257 | UN | 0.996 | 0.996 | 0.983 |
| SVHN | FROBINIT | W/O | 80M: 73257 | 80M | 0.993 | 0.993 | 0.987 |
| MEAN | FROBINIT | W/O | 80M: 73257 | | **0.987** | **0.987** | **0.938** |
| C10 | FROBINIT | W/O | 80M | C100 | 0.760 | 0.760 | 0.560 |
| C10 | FROBINIT | W/O | 80M | SVHN | 0.838 | 0.838 | 0.648 |
| C10 | FROBINIT | W/O | 80M | LFN | 0.900 | 0.900 | 0.820 |
| C10 | FROBINIT | W/O | 80M | UN | 0.985 | 0.985 | 0.945 |
| C10 | FROBINIT | W/O | 80M | 80M | 0.820 | 0.820 | 0.647 |
| MEAN | FROBINIT | W/O | 80M | | **0.861** | **0.861** | **0.724** |
| C10 | FROBINIT | W/O | 80M: 73257 | C100 | 0.776 | 0.776 | 0.644 |
| C10 | FROBINIT | W/O | 80M: 73257 | SVHN | 0.801 | 0.801 | 0.556 |
| C10 | FROBINIT | W/O | 80M: 73257 | LFN | 0.885 | 0.885 | 0.555 |
| C10 | FROBINIT | W/O | 80M: 73257 | UN | 0.869 | 0.869 | 0.849 |
| C10 | FROBINIT | W/O | 80M: 73257 | 80M | 0.838 | 0.838 | 0.699 |
| MEAN | FROBINIT | W/O | 80M: 73257 | | **0.834** | **0.834** | **0.661** |

Table 7: Performance of FROBInit w/ the Outlier Dataset -all data- and w/o using our $O(\mathbf{z})$.

| NORMAL | MODEL | $O(\mathbf{z})$ | OUTLIER | TEST | AUROC | AAUROC | GAUROC |
|--------|-------|-------|---------|------|-------|--------|--------|
| C10 | FROBINIT | W/O | SVHN | C100 | 0.829 | 0.829 | 0.153 |
| C10 | FROBINIT | W/O | SVHN | SVHN | **0.998** | **0.998** | **0.990** |
| C10 | FROBINIT | W/O | SVHN | LFN | 0.992 | 0.992 | 0.924 |
| MEAN | FROBINIT | W/O | SVHN | | **0.940** | **0.940** | **0.689** |
| C10 | FROBINIT | W/O | 80M: 73257 | C100 | 0.776 | 0.776 | 0.644 |
| C10 | FROBINIT | W/O | 80M: 73257 | SVHN | 0.801 | 0.801 | 0.556 |
| C10 | FROBINIT | W/O | 80M: 73257 | LFN | 0.885 | 0.885 | 0.555 |
| C10 | FROBINIT | W/O | 80M: 73257 | UN | 0.869 | 0.869 | 0.849 |
| C10 | FROBINIT | W/O | 80M: 73257 | 80M | 0.838 | 0.838 | 0.699 |
| MEAN | FROBINIT | W/O | 80M: 73257 | | **0.834** | **0.834** | **0.661** |

Table 8: OoD detection performance of FROB trained on the normal class with an OE set, using few-shots, and using boundary samples, $O(\mathbf{z})$, tested on different sets in AUROC, AAUROC, and GAUROC. We demonstrate FROB's efficacy and efficiency in Sec. 4.2. Here, "w/ $O$" means with our learned OoD, $O(\mathbf{z})$, while "w/o $O$" refers to without our self-generated anomalies, $O(\mathbf{z})$. *Here, C10 refers to CIFAR-10, C100 to CIFAR-100, UN to Uniform Noise, and LFN to Low Frequency Noise.*

| NORMAL | MODEL | $O(\mathbf{z})$ | OUTLIER, FS | TEST | AUROC | AAUROC | GAUROC |
|--------|-------|------|-------------|------|--------|---------|---------|
| C10 | FROB | w/ | SVHN: 1830 | C100 | 0.815 | 0.815 | 0.128 |
| C10 | FROB | w/ | SVHN: 1830 | SVHN | 0.997 | 0.997 | 0.990 |
| C10 | FROB | w/ | SVHN: 1830 | LFN | 0.985 | 0.985 | 0.642 |
| C10 | FROB | w/ | SVHN: 1830 | UN | 0.915 | 0.915 | 0.001 |
| C10 | FROB | w/ | SVHN: 915 | C100 | 0.834 | 0.834 | 0.124 |
| C10 | FROB | w/ | SVHN: 915 | SVHN | 0.995 | 0.995 | 0.984 |
| C10 | FROB | w/ | SVHN: 915 | LFN | 0.986 | 0.986 | 0.713 |
| C10 | FROB | w/ | SVHN: 915 | UN | 0.699 | 0.699 | 0.002 |
| C10 | FROB | w/ | SVHN: 732 | C100 | 0.829 | 0.829 | 0.100 |
| C10 | FROB | w/ | SVHN: 732 | SVHN | 0.995 | 0.995 | 0.981 |
| C10 | FROB | w/ | SVHN: 732 | LFN | 0.992 | 0.992 | 0.756 |
| C10 | FROB | w/ | SVHN: 732 | UN | 0.805 | 0.805 | 0.005 |
| C10 | FROB | w/ | SVHN: 457 | C100 | 0.827 | 0.827 | 0.107 |
| C10 | FROB | w/ | SVHN: 457 | SVHN | 0.997 | 0.997 | 0.982 |
| C10 | FROB | w/ | SVHN: 457 | LFN | 0.995 | 0.995 | 0.889 |
| C10 | FROB | w/ | SVHN: 457 | UN | 0.914 | 0.914 | 0.018 |
| C10 | FROB | w/ | SVHN: 100 | C100 | 0.846 | 0.846 | 0.060 |
| C10 | FROB | w/ | SVHN: 100 | SVHN | **0.996** | **0.996** | **0.950** |
| C10 | FROB | w/ | SVHN: 100 | LFN | 0.991 | 0.991 | 0.128 |
| C10 | FROB | w/ | SVHN: 100 | UN | 0.949 | 0.949 | 0.090 |
| C10 | FROB | w/ | SVHN: 80 | C100 | 0.837 | 0.837 | 0.070 |
| C10 | FROB | w/ | SVHN: 80 | SVHN | **0.995** | **0.995** | **0.928** |
| C10 | FROB | w/ | SVHN: 80 | LFN | 0.981 | 0.981 | 0.254 |
| C10 | FROB | w/ | SVHN: 80 | UN | 0.907 | 0.907 | 0.020 |
| C10 | FROB | w/ | SVHN: 0 | C100 | 0.865 | 0.865 | 0.003 |
| C10 | FROB | w/ | SVHN: 0 | SVHN | **0.895** | **0.895** | 0.023 |
| C10 | FROB | w/ | SVHN: 0 | LFN | 0.924 | 0.924 | 0.000 |
| C10 | FROB | w/ | SVHN: 0 | UN | 0.887 | 0.887 | 0.000 |

Table 9: OoD performance of FROB using an Outlier Set, using few-shots and our boundary, $O(\mathbf{z})$, tested on different sets in AUROC, AAUROC, and GAUROC, where "w/ $O$" means with our learned self-implicitly-generated OoD samples, $O(\mathbf{z})$. *In this Table, C10 refers to CIFAR-10, C100 to CIFAR-100, 80M to 80 Million Tiny Images, UN to Uniform Noise, and LFN to Low Frequency Noise.*

| NORMAL | MODEL | $O(\mathbf{z})$ | OUTLIER, FS | TEST DATA | AU ROC | AAU ROC | GAU ROC |
|--------|-------|------|-------------|-----------|--------|---------|---------|
| C10 | FROB | w/ | 80M: 73257, SVHN: 1830 | C100 | 0.850 | 0.850 | 0.585 |
| C10 | FROB | w/ | 80M: 73257, SVHN: 1830 | SVHN | 0.994 | 0.994 | 0.972 |
| C10 | FROB | w/ | 80M: 73257, SVHN: 1830 | LFN | 0.997 | 0.997 | 0.987 |
| C10 | FROB | w/ | 80M: 73257, SVHN: 1830 | UN | 0.967 | 0.967 | 0.865 |
| C10 | FROB | w/ | 80M: 73257, SVHN: 915 | C100 | 0.759 | 0.759 | 0.090 |
| C10 | FROB | w/ | 80M: 73257, SVHN: 915 | SVHN | 0.993 | 0.993 | 0.333 |
| C10 | FROB | w/ | 80M: 73257, SVHN: 915 | LFN | 0.993 | 0.993 | 0.040 |
| C10 | FROB | w/ | 80M: 73257, SVHN: 915 | UN | 0.434 | 0.434 | 0.010 |
| C10 | FROB | w/ | 80M: 73257, SVHN: 732 | C100 | 0.715 | 0.715 | 0.010 |
| C10 | FROB | w/ | 80M: 73257, SVHN: 732 | SVHN | 0.990 | 0.990 | 0.010 |
| C10 | FROB | w/ | 80M: 73257, SVHN: 732 | LFN | 0.869 | 0.869 | 0.010 |
| C10 | FROB | w/ | 80M: 73257, SVHN: 732 | UN | 0.988 | 0.988 | 0.010 |
| C10 | FROB | w/ | 80M: 73257, SVHN: 457 | C100 | 0.827 | 0.827 | 0.397 |
| C10 | FROB | w/ | 80M: 73257, SVHN: 457 | SVHN | 0.997 | 0.997 | 0.807 |
| C10 | FROB | w/ | 80M: 73257, SVHN: 457 | LFN | 0.997 | 0.997 | 0.841 |
| C10 | FROB | w/ | 80M: 73257, SVHN: 457 | UN | 0.914 | 0.914 | 0.842 |
| C10 | FROB | w/ | 80M: 73257, SVHN: 100 | C100 | 0.744 | 0.744 | 0.427 |
| C10 | FROB | w/ | 80M: 73257, SVHN: 100 | SVHN | **0.992** | **0.992** | **0.896** |
| C10 | FROB | w/ | 80M: 73257, SVHN: 100 | LFN | 0.985 | 0.985 | 0.912 |
| C10 | FROB | w/ | 80M: 73257, SVHN: 100 | UN | 0.934 | 0.934 | 0.911 |
| C10 | FROB | w/ | 80M: 73257, SVHN: 80 | C100 | 0.772 | 0.772 | 0.425 |
| C10 | FROB | w/ | 80M: 73257, SVHN: 80 | SVHN | **0.981** | **0.981** | **0.922** |
| C10 | FROB | w/ | 80M: 73257, SVHN: 80 | LFN | 0.990 | 0.990 | 0.951 |
| C10 | FROB | w/ | 80M: 73257, SVHN: 80 | UN | 0.901 | 0.901 | 0.788 |
| C10 | FROB | w/ | 80M: 73257, SVHN: 0 | C100 | 0.864 | 0.864 | 0.312 |
| C10 | FROB | w/ | 80M: 73257, SVHN: 0 | SVHN | **0.927** | **0.927** | **0.601** |
| C10 | FROB | w/ | 80M: 73257, SVHN: 0 | LFN | 0.891 | 0.891 | 0.301 |
| C10 | FROB | w/ | 80M: 73257, SVHN: 0 | UN | 0.865 | 0.865 | 0.212 |

Table 10: OoD performance of FROB with an Outlier Set, using $1830$ few-shots and our $O(\mathbf{z})$, tested on different sets in AUROC, AAUROC, and GAUROC. We compare to FROBInit w/o $O(\mathbf{z})$. *C10 refers to CIFAR-10, C100 to CIFAR-100, 80M to 80 Million Tiny Images, and UN to Uniform Noise.*

| NORMAL | $O(\mathbf{z})$ | OUTLIER | OUTLIER, FS | TEST | AUROC | AAUROC | GAUROC |
|---|---|---|---|---|---|---|---|
| C10 | w/ | NONE | SVHN: 1830 | C100 | 0.815 | 0.815 | 0.128 |
| C10 | w/ | NONE | SVHN: 1830 | SVHN | 0.997 | 0.997 | 0.990 |
| C10 | w/ | NONE | SVHN: 1830 | LFN | 0.985 | 0.985 | 0.642 |
| C10 | w/ | NONE | SVHN: 1830 | UN | 0.915 | 0.915 | 0.010 |
| C10 | w/o | NONE | SVHN: 1830 | C100 | 0.546 | 0.546 | 0.238 |
| C10 | w/o | NONE | SVHN: 1830 | SVHN | 0.847 | 0.847 | 0.728 |
| C10 | w/o | NONE | SVHN: 1830 | LFN | 0.742 | 0.742 | 0.528 |
| C10 | w/o | NONE | SVHN: 1830 | UN | 0.834 | 0.834 | 0.586 |
| C10 | w/ | 80M | SVHN: 1830 | C100 | 0.825 | 0.825 | **0.570** |
| C10 | w/ | 80M | SVHN: 1830 | SVHN | 0.994 | 0.994 | **0.980** |
| C10 | w/ | 80M | SVHN: 1830 | LFN | 0.995 | 0.995 | **0.992** |
| C10 | w/ | 80M | SVHN: 1830 | UN | 0.863 | 0.863 | **0.800** |
| C10 | w/o | 80M | SVHN: 1830 | C100 | 0.725 | 0.725 | 0.129 |
| C10 | w/o | 80M | SVHN: 1830 | SVHN | 0.983 | 0.983 | 0.970 |
| C10 | w/o | 80M | SVHN: 1830 | LFN | 0.985 | 0.985 | 0.699 |
| C10 | w/o | 80M | SVHN: 1830 | UN | 0.838 | 0.838 | 0.100 |

Table 11: Performance of FROB trained on normal CIFAR-10 using Outlier Set, using few-shots (FS) and learned boundary samples, $O(\mathbf{z})$, evaluated in AUROC, AAUROC, and GAUROC.

| NORMAL | $O(\mathbf{z})$ | OUTLIER | OUTLIER, FS | TEST | AUROC | AAUROC | GAUROC |
|---|---|---|---|---|---|---|---|
| C10 | w/ | NONE | SVHN: 915 | C100 | 0.834 | 0.834 | 0.124 |
| C10 | w/ | NONE | SVHN: 915 | SVHN | **0.995** | **0.995** | **0.984** |
| C10 | w/ | NONE | SVHN: 915 | LFN | 0.986 | 0.986 | 0.713 |
| C10 | w/ | NONE | SVHN: 915 | UN | 0.699 | 0.699 | 0.020 |
| C10 | w/o | NONE | SVHN: 915 | C100 | 0.512 | 0.512 | 0.209 |
| C10 | w/o | NONE | SVHN: 915 | SVHN | 0.824 | 0.824 | 0.706 |
| C10 | w/o | NONE | SVHN: 915 | LFN | 0.715 | 0.715 | 0.494 |
| C10 | w/o | NONE | SVHN: 915 | UN | 0.811 | 0.811 | 0.555 |
| C10 | w/ | 80M | SVHN: 915 | C100 | 0.744 | 0.744 | **0.413** |
| C10 | w/ | 80M | SVHN: 915 | SVHN | 0.988 | 0.988 | **0.969** |
| C10 | w/ | 80M | SVHN: 915 | LFN | 0.990 | 0.990 | **0.935** |
| C10 | w/ | 80M | SVHN: 915 | UN | 0.612 | 0.612 | **0.224** |
| C10 | w/o | 80M | SVHN: 915 | C100 | 0.722 | 0.722 | 0.554 |
| C10 | w/o | 80M | SVHN: 915 | SVHN | 0.988 | 0.988 | 0.969 |
| C10 | w/o | 80M | SVHN: 915 | LFN | 0.988 | 0.988 | 0.974 |
| C10 | w/o | 80M | SVHN: 915 | UN | 0.743 | 0.743 | 0.655 |

Table 12: OoD performance of FROB trained with $O(\mathbf{z})$, SVHN 100 few-shots, and Outlier Set.

| NORMAL | MODEL | $O(\mathbf{z})$ | OUTLIER, FS | TEST | AUROC | AAUROC | GAUROC |
|---|---|---|---|---|---|---|---|
| C10 | FROB | w/ | SVHN: 100 | C100 | 0.846 | 0.846 | 0.060 |
| C10 | FROB | w/ | SVHN: 100 | SVHN | **0.996** | **0.996** | **0.950** |
| C10 | FROB | w/ | SVHN: 100 | LFN | 0.991 | 0.991 | 0.128 |
| C10 | FROB | w/ | SVHN: 100 | UN | 0.949 | 0.949 | 0.090 |
| C10 | FROB | w/ | 80M:73257, SVHN:100 | C100 | 0.744 | 0.744 | **0.427** |
| C10 | FROB | w/ | 80M:73257, SVHN:100 | SVHN | 0.975 | 0.975 | **0.896** |
| C10 | FROB | w/ | 80M:73257, SVHN:100 | LFN | 0.985 | 0.985 | **0.912** |
| C10 | FROB | w/ | 80M:73257, SVHN:100 | UN | 0.934 | 0.934 | **0.911** |
| C10 | FROB | w/ | 80M:73257 | C100 | 0.846 | 0.846 | **0.010** |
| C10 | FROB | w/ | 80M: 73257 | SVHN | 0.872 | 0.872 | 0.050 |
| C10 | FROB | w/ | 80M:73257 | LFN | 0.900 | 0.900 | **0.005** |
| C10 | FROB | w/ | 80M:73257 | UN | 0.873 | 0.873 | **0.005** |
| C10 | FROB | w/ | C100 & SVHN:100 | C100 | 0.740 | 0.740 | **0.452** |
| C10 | FROB | w/ | C100 & SVHN:100 | SVHN | 0.978 | 0.978 | **0.913** |
| C10 | FROB | w/ | C100 & SVHN:100 | LFN | 0.990 | 0.990 | **0.959** |
| C10 | FROB | w/ | C100 & SVHN:100 | UN | 0.887 | 0.887 | **0.857** |

Table 13: OoD performance of FROB trained with $O(\mathbf{z})$, 80 SVHN few-shots, and Outlier Set.

| NORMAL | MODEL | $O(\mathbf{z})$ | OUTLIER, FS | TEST DATA | AUROC | AAUROC | GAUROC |
|---|---|---|---|---|---|---|---|
| C10 | FROB | w/ | SVHN: 80 | C100 | 0.837 | 0.837 | 0.070 |
| C10 | FROB | w/ | SVHN: 80 | SVHN | **0.995** | **0.995** | **0.928** |
| C10 | FROB | w/ | SVHN: 80 | LFN | 0.981 | 0.981 | 0.254 |
| C10 | FROB | w/ | SVHN: 80 | UN | 0.907 | 0.907 | 0.020 |
| C10 | FROB | w/ | 80M:73257, SVHN:80 | C100 | 0.765 | 0.765 | **0.345** |
| C10 | FROB | w/ | 80M:73257, SVHN:80 | SVHN | 0.981 | 0.981 | 0.904 |
| C10 | FROB | w/ | 80M:73257, SVHN:80 | LFN | 0.990 | 0.990 | **0.929** |
| C10 | FROB | w/ | 80M:73257, SVHN:80 | UN | 0.781 | 0.781 | **0.179** |
| C10 | FROB | w/ | 80M: 73257 | C100 | 0.846 | 0.846 | **0.010** |
| C10 | FROB | w/ | 80M: 73257 | SVHN | 0.872 | 0.872 | 0.050 |
| C10 | FROB | w/ | 80M: 73257 | LFN | 0.900 | 0.900 | **0.005** |
| C10 | FROB | w/ | 80M: 73257 | UN | 0.873 | 0.873 | **0.005** |

Table 14: OoD detection performance of FROB trained on normal SVHN using an Outlier Dataset, few-shot outliers, and the self-generated boundary samples, $O(\mathbf{z})$, evaluated on different sets in AUROC, AAUROC, and GAUROC. In this Table, "w/ $O$" means using the learned samples $O(\mathbf{z})$.

| NORMAL | MODEL | $O(\mathbf{z})$ | OUTLIER, FS | TEST | AUROC | AAUROC | GAUROC |
|--------|-------|------|-------------|------|-------|--------|--------|
| SVHN | FROB | w/ | C10: 600 | C100 | 0.991 | 0.991 | 0.956 |
| SVHN | FROB | w/ | C10: 600 | C10 | 0.996 | 0.996 | 0.982 |
| SVHN | FROB | w/ | C10: 600 | LFN | 0.914 | 0.914 | 0.381 |
| SVHN | FROB | w/ | C10: 600 | UN | 1.000 | 1.000 | 0.998 |
| SVHN | FROB | w/ | C10: 400 | C100 | 0.989 | 0.989 | 0.930 |
| SVHN | FROB | w/ | C10: 400 | C10 | **0.994** | **0.994** | **0.964** |
| SVHN | FROB | w/ | C10: 400 | LFN | 0.923 | 0.923 | 0.375 |
| SVHN | FROB | w/ | C10: 400 | UN | 0.997 | 0.997 | 0.978 |
| SVHN | FROB | w/ | C10: 200 | C100 | 0.990 | 0.990 | 0.941 |
| SVHN | FROB | w/ | C10: 200 | C10 | **0.996** | **0.996** | **0.967** |
| SVHN | FROB | w/ | C10: 200 | LFN | 0.947 | 0.947 | 0.573 |
| SVHN | FROB | w/ | C10: 200 | UN | 0.999 | 0.999 | 0.998 |
| SVHN | FROB | w/ | C10: 80 | C100 | 0.967 | 0.967 | 0.911 |
| SVHN | FROB | w/ | C10: 80 | C10 | **0.991** | **0.991** | **0.951** |
| SVHN | FROB | w/ | C10: 80 | LFN | 0.951 | 0.951 | 0.597 |
| SVHN | FROB | w/ | C10: 80 | UN | 0.989 | 0.989 | 0.967 |
| SVHN | FROB | w/ | C10: 0 | C100 | 0.951 | 0.951 | 0.000 |
| SVHN | FROB | w/ | C10: 0 | C10 | **0.958** | **0.958** | 0.000 |
| SVHN | FROB | w/ | C10: 0 | LFN | 0.961 | 0.961 | 0.000 |
| SVHN | FROB | w/ | C10: 0 | UN | 0.975 | 0.975 | 0.000 |

Table 15: OoD performance of FROB with the boundary, $O(\mathbf{z})$, in AUROC using One-Class Classification (OCC) and FS of 80 CIFAR-10 OCC anomalies. Comparison with benchmarks in this 80 few-shot setting (Sheynin et al., 2021). *FwODS refers to FROB with (w/) Outlier Dataset SVHN.*

| NORMAL | DROCC | GEOM | GOAD | HTD | SVDD | PASVDD | FROB | FwOES |
|--------|-------|------|------|-----|------|--------|------|-------|
| PLANE | 0.790 | 0.699 | 0.521 | 0.748 | 0.609 | 0.340 | **0.811** | **0.867** |
| CAR | 0.432 | 0.853 | 0.592 | **0.880** | 0.601 | 0.638 | 0.862 | 0.861 |
| BIRD | 0.682 | 0.608 | 0.507 | 0.624 | 0.446 | 0.400 | **0.721** | 0.707 |
| CAT | 0.557 | 0.629 | 0.538 | 0.601 | 0.587 | 0.549 | **0.748** | 0.787 |
| DEER | 0.572 | 0.563 | 0.627 | 0.501 | 0.563 | 0.500 | **0.742** | 0.727 |
| DOG | 0.644 | 0.765 | 0.525 | **0.784** | 0.609 | 0.482 | 0.771 | 0.782 |
| FROG | 0.509 | 0.699 | 0.515 | 0.753 | 0.585 | 0.570 | **0.826** | 0.884 |
| HORSE | 0.476 | 0.799 | 0.521 | **0.823** | 0.609 | 0.567 | 0.792 | 0.815 |
| SHIP | 0.770 | 0.840 | 0.704 | **0.874** | 0.748 | 0.440 | 0.826 | 0.792 |
| TRUCK | 0.424 | **0.834** | 0.697 | 0.812 | 0.721 | 0.612 | 0.744 | 0.799 |
| **MEAN** | 0.585 | 0.735 | 0.562 | 0.756 | 0.608 | 0.510 | **0.784** | **0.802** |

