# OpenReview forum: "FROB: Few-shot ROBust Model for Classification with Out-of-Distribution Detection"
_ICLR.cc/2022/Conference — ICLR 2022 Submitted_

### Official Review · Reviewer_94aV · 2021-10-18

**Correctness:** 3
**Technical Novelty And Significance:** 2
**Empirical Novelty And Significance:** 2
**Recommendation:** 5
**Confidence:** 2

**Main Review:**

This paper is generally well-written and well-structured. The experimental results are sufficient.

**Summary Of The Paper:**

This paper focuses on classification and Out-of-Distribution (OoD) detection in the few-shot setting. They propose the Few-shot ROBust (FROB) model for classification and few-shot OoD detection.

**Summary Of The Review:**

Generating low-confidence samples from random vectors is a common practice. All the terms in Eqn. (2), which is the key component of the proposed method, are trivial. The technical novelty and contribution of the proposed method is somewhat weak.

---

> ### Author Response · Authors · 2021-11-21
> **Reviewer 94aV**
>
> Reviewer 94aV. We would like to thank the reviewer for all his/her constructive and valuable comments. Equation numbers, page numbers, sections, and references refer to our paper, unless stated otherwise.
>
> >Generating low-confidence samples from random vectors is a common practice.
> All the terms in Eqn. (2), which is the key component of the proposed method, are trivial.
>
> The first term of our optimization (2) is the scattering loss, for dispersion to avoid mode collapse. The second term in (2) is the relative confidence loss to force the samples to the support boundary of the normal class distribution. The third term in our optimization (2) is the distance loss to penalize distance to normality using the distance from a point to a set. In (2), z are latent space samples from a standard Gaussian distribution, on page 2 in Sec. 2 (bottom of page 2). The optimization (2) is in accordance with the literature, for example (Lee et al., 2018a), (Dai et al., 2017), and (Dionelis et al., 2020b).
>
> >The technical novelty and contribution of the proposed method is somewhat weak.
>
> FROB is trained on a normal class set and generates the normal class distribution confidence boundary. These distribution boundary samples are set as OoD, and FROB performs negative training. One of the key takeaways and one of the main contributions of our paper is that FROB redesigns and streamlines the use of OoD Sets to work for few-shot samples, even for zero-shots, as FROB performs learned negative data augmentation to model the boundary of the support of the normal class distribution instead of using a large OoD Dataset, which constitutes an ad hoc choice of outliers and is used to model the complement of the support of the normal class distribution. The modified 80 Million Tiny Images OoD Dataset, when used to model and represent the entire complement of the support of the normal class distribution, leads to a feature-dependent ad hoc choice of outliers in Sec. 3 on pages 3 and 4.

---

### Official Review · Reviewer_qcc7 · 2021-10-31

**Correctness:** 3
**Technical Novelty And Significance:** 2
**Empirical Novelty And Significance:** 2
**Recommendation:** 3
**Confidence:** 5

**Main Review:**

The paper is generally well written and addresses an important issue of OoD detection combined with few-shot scenario.
The task is definitely very practical, of importance to the ML community, and with real world applications.

The proposed loss function. The only novel aspect would be the second term minimizing the softmax predictions for OoD samples. It would be good to compare it to existing, very similar loss functions in the OoD community such as [1, 2] where the intention is to penalize high softmax scores for OoD samples. What is the differentiating factor here?

Generation of adversarial examples. I am lacking here a deeper comparison to existing methods utilizing adversarial examples in a similar fashion (as close to border negatives) [3,4]. It is possible that the proposed approach is significantly different, however lack of comparisons and description does not allow to assess that.

In the reference section I am also missing a general discussion on adversarial methods and open-set approaches.

Since the authors pursue the few-shot OoD task, I believe it would be crucial to underline the difference from few-shot open-set classification/recognition (such as [2]). Quick scholar search shows multiple papers in the topic that I believe to be of high relevance here. They also establish benchmark datasets to be the datasets from few-shot classification such as miniImageNet, CUB or tinyImageNet rather than CIFAR-10 and SVHN which are good as a proof of concept, but require a more modern and difficult dataset to back them up.

The experiments and ablation studies seem interesting and well thought, however I am lacking some deeper explanation between non-obvious numbers present in the paper describing chosen amount of training data (e.g., 915, 1830, 73257). There might be an easy explanation there, but without it they seem quite random.

[1] Dhamija, Akshay Raj, Manuel Günther, and Terrance E. Boult. "Reducing network agnostophobia." Proceedings of the 32nd International Conference on Neural Information Processing Systems. 2018.

[2] Liu, Bo, et al. "Few-shot open-set recognition using meta-learning." Proceedings of the IEEE/CVF Conference on Computer Vision and Pattern Recognition. 2020.

[3] Kim, Jaehyung, Jongheon Jeong, and Jinwoo Shin. "M2m: Imbalanced classification via major-to-minor translation." Proceedings of the IEEE/CVF Conference on Computer Vision and Pattern Recognition. 2020.

[4] Kozerawski, Jedrzej, et al. "BLT: Balancing Long-Tailed Datasets with Adversarially-Perturbed Images." Proceedings of the Asian Conference on Computer Vision. 2020.

**Summary Of The Paper:**

The paper addresses an important issue of Out-of-Distribution detection in a few-shot setting. The authors propose to generate negative samples in an adversarial way to increase the OoD performance. Additionally they augment the loss function with additional term for OoD samples. They perform experiments of 2 benchmark datasets: CIFAR-10 and SVHN.

**Summary Of The Review:**

The paper touches interesting problem and proposes a valid solution, however lack of comparison with adversarial and few-shot open-set methods raises some questions as to novelty and quality of the experimental results. Additionally, experiments on more modern datasets should be performed, since CIFAR-10 and SVHN are quite simple.

Post-rebuttal: The lack of deeper discussion on related work, the lack of comparisons and modern datasets still causes me to keep my initial score. I believe more work needs to be done in order for it to be accepted.

---

> ### Author Response · Authors · 2021-11-21
> **Reviewer qcc7 (1 out of 3)**
>
> Reviewer qcc7 (1 out of 3). We would like to thank the reviewer for all his/her constructive and valuable comments. Equation numbers, page numbers, sections, and references refer to our paper, unless stated otherwise.
>
> >The proposed loss function. The only novel aspect would be the second term minimizing the softmax predictions for OoD samples.
>
> The first term of the proposed optimization task (2) is the scattering loss, for dispersion to avoid mode collapse. The second term in (2) is the relative confidence loss to force the samples to the support boundary of the normal class distribution. The third term in our optimization task (2) is the distance loss to penalize distance to normality using the distance from a point to a set. The optimization (2) is in accordance with the literature, e.g. (Lee et al., 2018a), (Dai et al., 2017), and (Dionelis et al., 2020b).
>
> >Compare it to existing, very similar loss functions such as [1, 2] where the intention is to penalize high softmax scores for OoD samples. What is the differentiating factor here?
> [1] Dhamija, Akshay Raj, Manuel Günther, and Terrance E. Boult. "Reducing network agnostophobia." Proceedings NeurIPS, 2018.
> [2] Liu, Bo, et al. "Few-shot open-set recognition using meta-learning." Proceedings CVPR, 2020.
>
> In Sec. 3 on pages 3 and 4, we present the models from the recent literature that most relate to FROB. The models that most relate to FROB are (Lee et al., 2018a), (Hein et al., 2019), (Bitterwolf et al., 2020), (Hendrycks et al., 2019), (Dai et al., 2017), and (Dionelis et al., 2020b). The modification procedure of removing the normal class from the 80 Million Tiny Images OoD Dataset leads to feature-dependent and dataset-dependent processes, as it is mentioned in the “Human prior” paragraph in Sec. 3 on page 4. On page 4, in the “Learned OoD samples” paragraph of Sec. 3, we clarify that FROB substantially differs from the Confidence-Calibrated Classifier (CCC) (Lee et al., 2018a) as: (i) CCC finds a threshold and not the boundary, (ii) CCC uses the OE set, U(y), where the labels follow a Uniform distribution, to compute this threshold and this is limiting as the threshold depends on U(y), which is an ad hoc choice of outliers, and (iii) FROB finds the confidence boundary and does not use U(y) to find this boundary.
> Regarding the Entropic Open-Set loss in [1], the three loss functions of the Entropic Open-Set, of the OE model in (Hendrycks et al., 2019), and of the Confidence Enhancing Data Augmentation (CEDA) model in (Hein et al., 2019) and (Bitterwolf et al., 2020), they all attain their global minimum when a Uniform probability distribution over the known classes is achieved. In our work, using FROB, we automate, learn, and mechanize the choice of the out-of-data-distribution samples, and this is the differentiating factor of FROB. The “Human prior” paragraph in Sec. 3 on page 4 of our paper clarifies this issue.
> Regarding the Open-Set Meta-Learning loss in [2], in the few-shot classification and learning setting, oPen sEt mEta LEaRning (PEELER) penalizes high softmax scores for the provided *given* OoD samples. In contrast, FROB learns the negative samples by performing sample generation on the normal class distribution boundary. The differentiating factor of FROB is this process of learning, automating, and mechanizing the selection of the OoD samples, and this is presented in Secs. 1-3 on pages 1-4.

---

> > ### Author Response · Authors · 2021-11-21
> > **Reviewer qcc7 (2 out of 3)**
> >
> > Reviewer qcc7 (2 out of 3).
> >
> > >Generation of adversarial samples. I am lacking here a deeper comparison to existing methods using adversarial samples in a similar fashion (as close to border negatives) [3,4].
> > [3] Kim, Jaehyung, Jongheon Jeong, and Jinwoo Shin. "M2m: Imbalanced classification via major-to-minor translation." Proceedings CVPR, 2020.
> > [4] Kozerawski, Jedrzej, et al. "BLT: Balancing Long-Tailed Datasets with Adversarially-Perturbed Images." Proceedings ACCV, 2020.
> >
> > The “Learned OoD samples” paragraph in Sec. 3 on page 4 compares FROB to existing methods using adversarial examples in a similar fashion, as close to border negatives. In addition, the “Human prior” paragraph in Sec. 3 on page 4 also compares FROB to existing techniques that use adversarial samples in a similar fashion, as close to border negatives. The proposed self-supervised boundary data augmentation obviates memorization, scalability, and data diversity problems arising from memory replay and prioritized experience replay (Zaheer et al., 2020; Pourreza et al., 2021).
> > Regarding the major-to-minor translation with respect to the decision boundary in [3], FROB substantially differs from [3]. The major-to-minor translation model in [3] proposes a solution for the high class imbalance problem while FROB solves the classification and few-shot OoD detection problem. FROB generates self-produced boundary samples and sets them as OoD data to more accurately perform few-shot OoD detection, as it is described in Sec. 2 on pages 2 and 3 of our paper. This learned boundary includes strong and adversarial OoD samples that are close to the normal class data distribution support and near high probability normal class samples. FROB includes optimal, relevant, useful, and task-specific anomalies in our training to robustly accurately detect few-shots of OoD data.
> > Regarding the Balancing Long-Tailed Datasets (BLT) in [4], FROB substantially differs from [4]. BLT tackles the high class imbalance problem with data augmentation. Instead, FROB solves the classification and few-shot OoD detection problem, and performs learned negative data augmentation.
> >
> > >In the references, I am missing a general discussion on adversarial methods and open-set approaches. Since the authors pursue the few-shot OoD task, I believe it would be crucial to underline the difference from few-shot open-set classification (e.g. [2]).
> >
> > By setting a threshold on the prediction confidence to perform OoD detection in addition to classification, using equation (3) in Sec. 2 on page 3, FROB performs Open-Set Recognition (OSR). OSR is the OoD detection capability of a classifier. OSR is the ability of discriminative models to know *when* they do not know. For K existing classes, the OSR classifier discriminates between the K+1 classes of OoD and the K base classes, i.e. K+1 classifier. Joint classification and OoD detection in high-dimensional spaces, images, is performed. Regarding the Open-Set Meta-Learning objective cost function in [2], in the few-shot classification and learning setting, the PEELER model penalizes high softmax scores for the provided/ given OoD samples. On the contrary, FROB learns the negative outlier samples by performing sample generation on the normal class data distribution boundary. The differentiating factor of FROB is this process of learning, automating, and mechanizing the selection of the OoD samples. This is presented in Secs. 1-3 on pages 1-4. Regarding adversarial methods and performing sample generation close to border negatives, FROB substantially differs from (Lee et al., 2018a), as it is detailed in Sec. 3 on page 4, in the “Learned OoD samples” paragraph. FROB also substantially differs from (Hein et al., 2019), (Bitterwolf et al., 2020), (Hendrycks et al., 2019), (Dai et al., 2017), and (Dionelis et al., 2020b).

---

> > > ### Author Response · Authors · 2021-11-21
> > > **Reviewer qcc7 (3 out of 3)**
> > >
> > > Reviewer qcc7 (3 out of 3).
> > >
> > > >They also establish benchmark datasets to be the datasets from few-shot classification such as miniImageNet, CUB or tinyImageNet rather than CIFAR-10 and SVHN which are good as a proof of concept, but require more modern difficult dataset to back them up.
> > >
> > > We would like to thank the reviewer for this comment. The section Evaluation and Results, i.e. Sec. 4, is from page 4 to page 9 and starts from evaluating FROB without the learned boundary and comparing it to benchmarks in Sec. 4.1 on pages 5 and 6. Next, the section Evaluation and Results evaluates FROB with the generated boundary samples and compares it to FROB without the learned boundary in Sec. 4.2 on pages 7 and 8. Then, this Evaluation and Results section evaluates FROB with the learned boundary samples using unseen, in the wild, datasets in Sec. 4.3 on pages 8 and 9. Finally, the section Evaluation and Results evaluates FROB with the generated boundary on OCC CIFAR-10 data in Sec. 4.4 on page 9. Hence, to compare with the benchmark models in Sec. 4.1 on pages 5 and 6, we used the CIFAR-10 and SVHN normal class benchmark datasets in Sec. 4 on pages 4-9.
> > >
> > > >I am lacking some deeper explanation between non-obvious numbers present in the paper describing chosen amount of training data (e.g., 915, 1830, 73257). There might be an easy explanation there, but without it they seem quite random.
> > >
> > > SVHN has 73257 training samples (Netzer et al., 2011) (http://ufldl.stanford.edu/housenumbers/). This is the only reason that 73257 80M samples were chosen. On page 4 in Sec. 4, it is mentioned that “To examine the robustness to the number of few-shot samples, we decrease the number of few-shots by dividing them by two”. The outlier few-shot *data* are selected by performing random sampling.
> > >
> > > >The paper touches an interesting problem and proposes a valid solution, however lack of comparison with adversarial and few-shot open-set methods raises some questions as to novelty and quality of the experimental results. Additionally, experiments on more modern datasets should be performed, since CIFAR-10 and SVHN are quite simple.
> > >
> > > To compare with the benchmark models in Sec. 4.1 on pages 5 and 6, we used the CIFAR-10 and SVHN normal class benchmark datasets in Sec. 4 on pages 4-9.

---

> > > > ### Comment · Reviewer_qcc7 · 2021-11-30
> > > > **Post-rebuttal**
> > > >
> > > > Thank you for the elaborate answers here. I think my main concerns are still here which is the lack of deeper discussion on related work, the lack of comparisons and modern datasets still causes me to keep my initial score.

---

### Official Review · Reviewer_1PDU · 2021-11-02

**Correctness:** 2
**Technical Novelty And Significance:** 3
**Empirical Novelty And Significance:** 1
**Recommendation:** 3
**Confidence:** 3

**Main Review:**

Strengths: the proposed method provides an improved version of outlier exposure (OE) method, by combining the self-generated boundary with the general outliers like 80M tiny images.

Weakness:
1. The main method is not clear to me, in particular,

In Eq (1), it is not clear which class is the f(Z_m) in the second term refers to.
In Eq (2), it is not explained what are i and j.

2. The results are not clearly presented.

In Figure 2, the curve for CIFAR-100 starts from a AUROC number below 0.5, i.e. when there is no outlier exposure, the model trained on CIFAR10 cannot distinguish between CIFAR-10 test and CIFAR-100 test. That’s strange because the current SOTA AUROC based on a wide resnet model is already around 0.95 [1]. Similarly, the current SOTA AUROC for SVHN is at least 0.91 (see Table 7 in [2]), while the value indicated in Figure 2 is only around 0.6.

[1] Hongjie Zhang, Ang Li, Jie Guo, and Yanwen Guo. Hybrid models for open set recognition. European Conference on Computer Vision, 2020.
[2] Hendrycks, Dan, Mantas Mazeika, and Thomas Dietterich. "Deep anomaly detection with outlier exposure." arXiv preprint arXiv:1812.04606 (2018).)

3. The usage of the term few-shot is not well suited.

The paper claimed their method is for few-shot OE. But the results show that "The FROB performance decreases with reducing number of SVHN fewshots. A small AUROC of 0.5 is reached for approximately 800 few-shots for CIFAR-100..",  “for few-shots less than 1800, the modeling error of OE covering the full complement of the support of the normal class is high”. In my opinion, 800/ 1800 images for training should no longer be called as few-shot.

4. The conclusion needs to be adjusted.

I found the mean AUROC value in Table 4 for FROB was computed wrongly. It seems that OE, CEDA, and ACET all have higher mean AUROC than FROB. Also, in particular for near-OOD benchmark CIFAR-10 vs CIFAR-100, FROB only has 0.76 AUROC, much lower compared with 0.924 for OE.

5. The effect of adding the generated boundary points O(Z) is not clear.

In Table 2, it would be good if you can provide OCC {FROB, FROB W/ OE SVHN} {w/ O(z), w/o O(z)} for direct study the effect of OE and O(z).



**Summary Of The Paper:**

The paper proposes to generate the support boundary of the normal class distribution and combine it with few-shot Outlier Exposure (OE) to improve the OOD detection performance. The proposed method is evaluated at multiple benchmark datasets, and is compared with a few baseline methods.

**Summary Of The Review:**

The paper proposes an interesting method for combining generated OOD that is customized to the training data with the general natural outlier images for outlier exposure. However, the paper needs to be further polished such that the method and results can be clearly presented.

---

> ### Author Response · Authors · 2021-11-21
> **Reviewer 1PDU (1 out of 3)**
>
> Reviewer 1PDU (1 out of 3). We would like to thank the reviewer for all his/her constructive and valuable comments. Equation numbers, page numbers, sections, and references refer to our paper, unless stated otherwise.
>
> >1. The main method is not clear to me. In Eq (1), it is not clear which class is the f(Z_m) in the second term refers to. In Eq (2), it is not explained what are i and j.
>
> FROB performs learned negative data augmentation to model the boundary of the support of the normal class distribution rather than using a large general OoD Set, e.g. 80 Million Tiny Images or SVHN or CIFAR-100, which constitutes an ad hoc choice of outliers and is used to model the complement of the support of the normal class distribution. Modelling the boundary of the support of the normal class data distribution needs less samples than modelling and representing the entire full complement of the support of the normal class distribution. The boundary provides robustness to the number of few-shots for OoD detection. Optimization (1) is described in Sec. 2 on pages 2 and 3. It is in accordance with the literature, e.g. (Hein et al., 2019) and (Bitterwolf et al., 2020): equations (1)-(3) in (Hein et al., 2019).
> In (2), z are latent space samples from a standard Gaussian distribution, as it is stated on page 2 in Sec. 2 (i.e. at the bottom of page 2). Thus, z_j is the j-th sample in the latent space from a standard Gaussian distribution, and z_i is the i-th sample in the latent space from a standard Gaussian distribution.
>
> >In Fig. 2, CIFAR-100 starts from AUROC below 0.5, i.e. when there is no outlier exposure, the model trained on CIFAR10 cannot distinguish between CIFAR-10 and CIFAR-100.
>
> Fig. 2 shows the OoD detection performance of FROBInit, where we denote FROB without the learned boundary and without few-shots by FROBInit, in an initial ablation study. Fig. 4 shows the OoD performance of our proposed FROB model. When the normal class is CIFAR-10 and the test set is CIFAR-100, then FROB with the learned boundary achieves an AUROC of 0.865 for zero-shots (Table 8). The model in (Bitterwolf et al., 2021) yields an AUROC of 0.843 (Table 1 in (Bitterwolf et al., 2021)) and the model in (Hein et al., 2019) achieves an AUROC of 0.856 (Table 1 in (Hein et al., 2019)). The OE model achieves an AUROC of 0.879 for zero-shots (Table 7 in (Hendrycks et al., 2019)).
> The section Evaluation and Results, i.e. Sec. 4, is from page 4 to page 9 and starts from evaluating FROB *without the boundary* and comparing it to benchmarks in Sec. 4.1 on pages 5 and 6. Next, the section Evaluation and Results evaluates FROB with the boundary and compares it to FROB without the learned boundary in Sec. 4.2 on pages 7 and 8. Then, this Evaluation section evaluates FROB with the learned boundary using unseen, in the wild, datasets in Sec. 4.3 on pages 8 and 9. Finally, the section Evaluation and Results evaluates FROB with the generated boundary on OCC CIFAR-10 data in Sec. 4.4 on page 9. Hence, Sec. 4.1 refers to the evaluation of FROB without the learned boundary.
>
> >That’s strange as the current SOTA AUROC with a wide resnet is around 0.95 [1].
> [1] Hongjie Zhang, Ang Li, Jie Guo, and Yanwen Guo. Hybrid models for open set recognition. European Conference on Computer Vision, 2020.
>
> Fig. 4 shows the performance of FROB, while Fig. 2 shows the OoD detection performance of FROBInit. FROB refers to FROB with the boundary, while FROBInit refers to FROB *without* the boundary. FROB achieves AUROC 0.865 for normal class CIFAR-10 and test set CIFAR-100. “Hybrid models for open set recognition” performs joint classification and explicit probability density estimation using a flow invertible generative model. In our work, we do not use explicit probability density estimation with *invertible* generators. Instead, we use discriminative classifiers. We aim at improving all the AUROC, AAUROC, and GAUROC metrics. We examine and are interested in the robustness of our model and in the worst-case OoD detection performance using l_infinity-norm perturbations for each of the OoD samples. In contrast, [1] examines only the AUROC. It does not report AAUROC and GAUROC. FROB substantially differs from [1] as FROB addresses the overconfident predictions problem of classifiers, sets a threshold on the confidence for OoD detection, performs learned negative data augmentation by generating the boundary, and redesigns the use of OoD Sets to work for few-shots, even for zero-shots. “Hybrid models for open set recognition” does not perform and evaluate few-shot OoD detection.

---

> > ### Author Response · Authors · 2021-11-21
> > **Reviewer 1PDU (2 out of 3)**
> >
> > Reviewer 1PDU (2 out of 3).
> >
> > >Similarly, the current SOTA AUROC for SVHN is at least 0.91 (see Table 7 in (Hendrycks et al., 2019)), while the value indicated in Fig. 2 is only around 0.6.
> >
> > Fig. 2 shows the OoD detection performance of FROBInit, where we denote FROB *without* the boundary and without few-shots by FROBInit, in an initial ablation study. Fig. 4 presents the AUROC of our proposed FROB model. When the normal class is CIFAR-10 and the test dataset is SVHN, then FROB *with* the boundary yields AUROC 0.895 for zero-shots (Table 8). The CCC model in (Lee et al., 2018a) yields AUROC 0.466 for zero-shots (Table 1 in (Lee et al., 2018a)). Moreover, for this setting, the model proposed in (Hein et al., 2019) yields an AUROC of 0.850 (Table 1 in (Hein et al., 2019)). For this setting, the OE model yields AUROC 0.918 for zero-shots (Table 7 in (Hendrycks et al., 2019)).
> > The AUROC for FROB with the boundary is approximately 0.9 in Fig. 4 on page 7. Table 1 in (Lee et al., 2018a) reports AUROC 0.47 for this setting. This is highly architecture-dependent. Table 1 in (Hein et al., 2019) reports AUROC 0.85 for this setting, and we again note that this is architecture-dependent.
> > For normal class CIFAR-10 and test set CIFAR-100, FROB with the boundary achieves AUROC 0.865 for zero-shots (Table 8). For this setting, (Bitterwolf et al., 2021) yields AUROC 0.843 (Table 1 in (Bitterwolf et al., 2021)) and (Hein et al., 2019) achieves AUROC 0.856 (Table 1 in (Hein et al., 2019)). The OE model achieves an AUROC of 0.879 for zero-shots (Table 7 in (Hendrycks et al., 2019)).
> > Furthermore, for normal class SVHN and test set CIFAR-10, FROB with the boundary achieves AUROC 0.958 for zero-shots (Table 14). The model in (Lee et al., 2018a) achieves AUROC 0.626 for zero-shots (Table 1 in (Lee et al., 2018a)). The model in (Hein et al., 2019) achieves AUROC 0.938 (Table 1 in (Hein et al., 2019)). The OE model achieves AUROC 0.980 for zero-shots (Table 7 in (Hendrycks et al., 2019)).
> > For normal class SVHN and test set *CIFAR-100*, FROB with the boundary yields AUROC 0.951 for zero-shots (Table 14). (Hein et al., 2019) yields AUROC 0.935 (Table 1 in (Hein et al., 2019)).
> >
> > >3. The term few-shot is not well suited. The paper claimed it is for few-shot OE. "FROB performance decreases with reducing number of SVHN few-shots.
> >
> > FROB is for classification and *few-shot* OoD detection, and uses our self-supervised learning boundary. In the few-shot setting, we evaluate FROB and its robustness by reducing the number of few-shots. In our Tables, FS-O(z) denotes that FROB models the boundary of the support of the normal class distribution rather than modelling the entire full complement of the support of the normal class distribution. We model the boundary instead of modelling and representing the complement of the normal class support. In our Tables, FS-OE denotes few-shot samples from an OoD Set, SVHN or CIFAR-10, in *reduced* number. OE DATA denotes samples from an Outlier Set, 80 Million Tiny Images.
> > FROB performs classification and few-shot OoD detection using self-produced samples on the distribution confidence boundary. It is evaluated in Figs. 4 and 5. Fig. 2 evaluates FROBInit which is *without* the learned boundary samples. This is an initial ablation study, i.e. FROBInit. As the number of few-shots decreases, in Fig. 4, we observe a small decrease in the AUROC OoD detection performance of FROB with the generated boundary. Comparing Fig. 4 to Fig. 2, we observe that FROB with the O(z) samples is effective and leads to robustness to the number of few-shots, compared to FROBInit.
> > In Sec. 4.1.2 on page 5, it is mentioned that “The FROB performance decreases with reducing number of SVHN few- shots.”, and this sentence refers to FROB without (w/o) the learned boundary samples.
> > The sentence from our paper that the reviewer refers to is for the evaluation of FROB without the learned boundary. The section Evaluation and Results, i.e. Sec. 4, is from page 4 to page 9 and starts from evaluating FROB without the boundary and comparing it to benchmarks in Sec. 4.1 on pages 5 and 6. Next, the section Evaluation and Results evaluates FROB with the boundary and compares it to FROB without the boundary in Sec. 4.2 on pages 7 and 8. Then, this Evaluation section evaluates FROB with the boundary using unseen, in the wild, datasets in Sec. 4.3 on pages 8 and 9. Finally, the Evaluation section evaluates FROB with the boundary on OCC CIFAR-10 data in Sec. 4.4 on page 9. Based on this explanation, we again note that the sentence “The FROB performance decreases with reducing number of SVHN few-shots.” is in Sec. 4.1 and refers to FROB without (w/o) the self-learned boundary.

---

> > > ### Author Response · Authors · 2021-11-21
> > > **Reviewer 1PDU (3 out of 3)**
> > >
> > > Reviewer 1PDU (3 out of 3).
> > >
> > > >A small AUROC of 0.5 is reached for approximately 800 few-shots for CIFAR-100..", “for few-shots less than 1800, the modeling error of OE covering the full complement of the support of the normal class is high”. 800 images for training should no longer be called few-shot.
> > >
> > > We agree that 800 are samples and not few-shots. Fig. 2 refers to FROBInit. Figs. 4 and 5 refer to FROB and evaluate our model for few-shot OoD detection, by decreasing the number of few-shots until zero.
> > >
> > > >I found the mean AUROC value in Table 4 for FROB was computed wrongly. It seems that OE, CEDA, and ACET all have higher mean AUROC than FROB.
> > >
> > > We would like to thank the reviewer for this typo in Table 4. We agree that in Table 4 the OE model, CEDA, and ACET outperform FROBInit in AUROC, where we denote FROB *without* the learned boundary and without using few-shots by FROBInit. FROBInit in Table 4 outperforms the models GOOD and CCC. According to Table 1, FROB outperforms the OE, CEDA, CCC, and GOOD models in AAUROC and GAUROC on average. Tables 1, 3, and 4, compare FROB without the boundary and without using few-shots, i.e. FROBInit, to benchmarks, in an initial ablation study. We have now clarified this; we emphasize the cases where FROB without the learned boundary is examined in an initial ablation study. Tables 8 to 14 present the OoD detection performance of FROB which integrates the self-generated confidence boundary, the few-shots, and eventually the outliers. Tables 2 and 15 show the OoD detection performance of FROB which integrates the self-generated boundary, the few-shot samples, and eventually the outliers, evaluated using OCC. Table 2 compares FROB to the benchmarks in (Sheynin et al., 2021) using the OCC evaluation for 80 few-shots. Tables 2 and 15 show the OCC evaluation of FROB compared to benchmarks GOAD, GEOM, DROCC, SVDD and PaSVDD.
> > >
> > > >Also, in particular for near-OOD benchmark CIFAR-10 vs CIFAR-100, FROB only has 0.76 AUROC, much lower compared with 0.924 for OE.
> > >
> > > We would like to thank the reviewer for his/her comment about Table 4. Table 4, as well as Table 1 and Sec. 4.1, refer to the evaluation of FROB without the boundary, to the initial FROBInit ablation study.
> > >
> > > >In Table 2, it would be good if you can provide OCC {FROB, FROB w/ OE SVHN} {w/ O(z), w/o O(z)} for direct study the effect of OE and O(z).
> > >
> > > When the normal class is CIFAR-10 or OCC CIFAR-10, using SVHN as the OoD Dataset is the most beneficial, then using 80 Million Tiny Images, and then CIFAR-100. Regarding Few-Shot OE (FS-OE), using the boundary is beneficial and effective. FS-OE is used to denote that FROB models the boundary of the support of the normal class distribution instead of modelling and representing the entire full complement of the support of the normal class distribution. FROB w/ O(z) is better than FROB w/o O(z). For few-shot OoD detection, FROB w/ O(z) w/ OE SVHN is better than FROB w/ O(z) w/o OE SVHN, which is better than FROB w/o O(z) w/ OE SVHN, which is better than FROB w/o O(z) w/o OE SVHN.

---

### Official Review · Reviewer_r838 · 2021-11-02

**Correctness:** 2
**Technical Novelty And Significance:** 3
**Empirical Novelty And Significance:** 2
**Recommendation:** 3
**Confidence:** 4

**Details Of Ethics Concerns:**

This paper heavily uses the retracted 80M tiny images dataset.

**Main Review:**

Strengths:
 - The proposed method achieves new SOTA results for OOD detection
 - The use of a generator for class boundary confidence training is novel/interesting

Weaknesses:
 - The paper is quite difficult to read. Many lines are repeated in slightly varied ways; 20% of the lines could be removed to enhance clarity and to be more concise.
 - The proposed method is not well described in the main text. Specifically, equation 1 appears incorrect, as the numerator in term 2 is a vector of class logits "f(Z_m)". The model and training setup for O(z) is hard to follow and also appears incorrect as well. Equation 2 term 1 has "j!=i" yet i does not appear in the equation. z_j is not defined, is it the feature representation of the j-th outlier? The intuition of terms 2 and 3 is unclear, more details on exactly what each of the 3 terms are providing to the O model would increase clarity significantly.
 - Two of the primary metrics, AAUROC and GAUROC, are not defined. Both of those metrics are designed around adversarial perturbations, which is not clear how it applies to this work.
- Table 1 has misreported metrics. Cifar10 with OE outperforms FROB on each individual dataset when looking at table 4 in the appendix.
- The outlier dataset size appears to be randomly selected/cherry picked. How did the authors choose the 1830 SVHN examples or 73257 80M examples?
- The final results of using various Outlier datasets are very strange. The results in table 12, C10 vs. C100 are non-intuitive to me. Why is it that using 100 SVHN examples gives the exact same AUC as 70k examples from 80M?
- The main discussion of the results has several pointers to large appendix tables, making comparison of results difficult and must remind the authors appendix reading is supposed to be optional, where the main text is self-supported.
- Finally, the overall results are not a fair comparison between methods. Specifically, how does OE perform when it only accesses the same number of outlier examples as FROB? Without data like this, I cannot evaluate the efficacy of FROB.

Additionally, the paper often uses 80M tiny images dataset, which has been retracted by the original creators.

Post rebuttal:
I do not feel as though many of my concerns have been addressed, and I will be keeping my original score.
- I still don't understand the model and training setup for O(z). I do not even know if O(z) is an MLP or CNN.
- The focus on AAUROC and GAUROC is misplaced. Hein's work with these metrics is somewhat related, but I do not think they are appropriate for evaluation with FROB.
- Too much information is being used from the appendix as primary reference material
- It still seems normal outlier exposure is a better choice on average than FROB, that is, I am still not sure what the guidelines are for choosing FROB over OE.
- 80 Million Tiny Images is inappropriate to develop new algorithms with. Other conferences (NuerIPS) will desk reject any papers using this dataset.

**Summary Of The Paper:**

This paper introduces a Few-shot OOD classification model, named FROB. Using self-supervised learning and generative model for enhanced confidence on class boundary, they achieve SOTA results compared to other similar OOD detection methods.

**Summary Of The Review:**

While this paper offers an interesting angle into OOD detection with few shot examples, several flaws regarding writing/methodology/results must be addressed before this work is ready for publication.

---

> ### Author Response · Authors · 2021-11-21
> **Reviewer r838 (1 out of 3)**
>
> Reviewer r838 (1 out of 3). We would like to thank the reviewer for all his/her constructive and valuable comments. Equation numbers, page numbers, sections, and references refer to our paper, unless stated otherwise.
>
> >Equation 1 appears incorrect, as the numerator in term 2 is a vector of class logits "f(Z_m)".
>
> The exp(.) operation has as input a vector and as output a vector. The exp(.) operation is performed for every element of the vector. The optimization (1) is in line with the literature, e.g. (Hein et al., 2019) and (Bitterwolf et al., 2020). Equations (1)-(3) in (Hein et al., 2019) clarify this issue concerning exp(.).
>
> >The model and training setup for O(z) is hard to follow and appears incorrect. The intuition of terms 2 and 3 is unclear, more details on what each of the 3 terms provide. z_j is not defined, is it the feature representation of the j-th outlier?
>
> FROB performs self-learned negative data augmentation to model the boundary of the support of the normal class distribution. The first term of our optimization (2) is the scattering loss, for dispersion to avoid mode collapse and achieve diversity and variation. The second term in (2) is the relative confidence loss to push the samples OoD. Hence, FROB forces the samples to the support boundary of the normal class distribution. The third term in our optimization (2) is the distance loss to penalize distance to normality using the distance from a point to a set. Hence, the samples go to the distribution confidence boundary. We define the confidence boundary when the second and third terms are zero, and minimize our loss (2) using Gradient Descent. Our optimization task (2) is in line with the literature, for example (Lee et al., 2018a), (Dai et al., 2017), and (Dionelis et al., 2020b).
> In (2), z are latent space samples from a standard Gaussian distribution, on page 2 in Sec. 2 (bottom of page 2). Hence, z_j is the j-th sample in the latent space from a standard Gaussian distribution.
>
> >AAUROC and GAUROC are not defined. Both of those metrics are designed around adversarial perturbations, which is not clear how it applies to this work.
>
> AAUROC and GAUROC are defined in Sec. 4 on page 4 using (Bitterwolf et al., 2020) and (Croce & Hein, 2020). The AAUROC focuses on the worst-case OoD detection performance using l_infinity-norm perturbations for the OoD samples. It uses the maximal confidence in the l_infinity-norm ball around each OoD image. The AAUROC finds a *lower* bound on the maximal confidence in the l_infinity-norm ball and the image domain. These worst-case confidences for the OoD are used for the AUROC.
> The GAUROC also focuses on the worst-case OoD detection performance using l_infinity-norm perturbations for the OoD samples. It uses the maximal confidence in the l_infinity-norm ball around each OoD image. It finds an *upper* bound on the maximal confidence in the l_infinity-norm ball and the image domain. The guaranteed confidences for the OoD are then used for the AUROC.
> AAUROC and GAUROC are suitable metrics for evaluating the robustness of OoD detection models focusing on the worst-case OoD detection performance using l_infinity-norm perturbations for each of the OoD samples. To strengthen the robustness evaluation of FROB and compare to benchmarks in Table 1, Figs. 2-7, and the Tables in the Appendix, in addition to AUROC, we also evaluate FROB with AAUROC and GAUROC. In Secs. 4.1-4.3, all the AUROC, AAUROC, and GAUROC metrics are examined. Both AAUROC and GAUROC are suitable for evaluating few-shot OoD detection models, including FROB, and also OoD detection models, including GOOD (Bitterwolf et al., 2020) and FROB.
>
> >Table 1 has misreported metrics. Cifar10 with Outlier Exposure outperforms FROB on each individual dataset when looking at Table 4 in the Appendix.
>
> We would like to thank the reviewer for this typo in Table 1. We agree that in Table 4, the OE model outperforms FROBInit in AUROC, where we denote FROB without the learned boundary and without using few-shots by FROBInit. FROBInit in Table 4 outperforms the OE model in AAUROC and GAUROC. Tables 1, 3, and 4, compare initial FROBInit without the boundary and without using few-shots, i.e. FROBInit, with benchmarks, in an *initial* ablation study. We have clarified this in our paper; we have emphasized the cases where FROB without the boundary, FROBInit, is examined in an initial ablation study. Tables 8 to 14 show the OoD detection performance of FROB which integrates the self-generated boundary, the few-shots, and the outliers. Tables 2 and 15 show the OoD detection performance of FROB evaluated using OCC. Table 2 compares FROB with the benchmarks in (Sheynin et al., 2021) for 80 few-shots. Tables 2 and 15 show the OCC evaluation of FROB compared to benchmarks in (Sheynin et al., 2021): GOAD, GEOM, DROCC, SVDD and PaSVDD models.

---

> > ### Author Response · Authors · 2021-11-21
> > **Reviewer r838 (2 out of 3)**
> >
> > Reviewer r838 (2 out of 3).
> >
> > >The outlier dataset size appears to be randomly selected/cherry picked. How did the authors choose the 1830 SVHN examples or 73257 80M examples?
> >
> > The SVHN dataset has 73257 training samples (Netzer et al., 2011). This can also be found in (http://ufldl.stanford.edu/housenumbers/). This is the only reason that 73257 80M samples were chosen. On page 4 in Sec. 4, we mention our evaluation strategy: “To examine the robustness to the number of few-shot samples, we decrease the number of few-shots by dividing them by two”. We hence divide the number of few-shot samples by 2 in our robustness evaluation analysis.
> >
> > >The results of using various Outlier Sets are very strange. The results in table 12, C10 vs. C100 are non-intuitive. Why does using 100 SVHN samples give the same AUC as 70k 80M samples?
> >
> > Table 12 shows that using an OoD Dataset, e.g. CIFAR-100 or 80 Million Tiny Images 73257 samples, leads to improved GAUROC, in Sec. 4.3.1. We observe that there is no need to use an OoD Set when the few-shots and the test samples originate from the same set, e.g. SVHN. In the last group of results in Table 12, we observe that using CIFAR-100 as the OoD Set is the least beneficial in AUROC. Assessing the evaluation results vis-a-vis our methodology, we observe that FROB generates the self-produced distribution boundary in a learned negative data augmentation manner and this leads to robustness to the number of few-shots. FROB performs few-shot OoD detection and achieves robustness.
> > For normal class CIFAR-10 in Table 12, for FROB with the boundary, we observe that when using 100 few-shots from SVHN (first group of four lines), we obtain improved AUC values compared to using 70k of 80M (third group of four lines), for almost all the test sets. Observing Fig. 4, when the normal class is CIFAR-10, we observe that we continue to obtain improved high AUROC values even with a decreased number of SVHN samples till zero, when testing on SVHN and Low-frequency noise datasets.
> > A similar behavior is shown in Fig. 5, when the normal class is SVHN, we continue to obtain improved high AUROC even with a decreased number of samples till zero of few-shots from CIFAR-10, for testing on CIFAR-10, CIFAR-100 and Low-frequency noise sets. This behavior is related to the generated samples on the normal class boundary and to the training of the classifier which, using our algorithm, learns to move away from the OoD samples. FROB succeeds in maintaining this OoD detection capability up to a very small number of few-shot samples, till zero-shots. We observe that independent of the benchmark training set, when we provide few-shots from a cross-domain dataset and test on the same set, we obtain high performance. FROB is effective/ successful when we test on a new unknown set. Comparing the first two groups of results in Table 12, we observe that using a general OoD Dataset in a few-shot setting is not beneficial in AUROC and AAUROC metrics, but is beneficial in GAUROC.
> > Using 100 SVHN samples gives comparable AUC results to using 70k 80 Million Tiny Images samples, in Table 12, when the normal class is CIFAR-10, because of the beneficial effect of the boundary of FROB. SVHN is a good OoD Dataset when the normal class is CIFAR-10. By including the boundary, we obtain comparable AUC results when using 100 SVHN samples compared to using 70k 80 Million Tiny Images samples. When operating under a fixed budget and constrained sampling complexity for the OoD samples, using 100 few-shots is more efficient than using 70000 samples.

---

> > > ### Author Response · Authors · 2021-11-21
> > > **Reviewer r838 (3 out of 3)**
> > >
> > > Reviewer r838 (3 out of 3).
> > >
> > > >The overall results are not a fair comparison between methods. How does OE perform when it only accesses the same number of outlier examples as FROB?
> > >
> > > For normal class CIFAR-10 and test set CIFAR-100, FROB with the learned boundary achieves an AUROC of 0.865 for zero-shots (Table 8). For this setting, (Bitterwolf et al., 2021) yields an AUROC of 0.843 (Table 1 in (Bitterwolf et al., 2021)) and (Hein et al., 2019) achieves an AUROC of 0.856 (Table 1 in (Hein et al., 2019)). The OE model achieves an AUROC of 0.879 for zero-shots (Table 7 in (Hendrycks et al., 2019)). This leads to a difference of 0.014, i.e. 1.6%, compared to FROB with the boundary.
> > > For normal class CIFAR-10 and test set SVHN, FROB *with* the boundary yields an AUROC of 0.895 for zero-shots (Table 8 in our paper). For this setting, the CCC model proposed in (Lee et al., 2018a) yields an AUROC of 0.466 for zero-shots (Table 1 in (Lee et al., 2018a)). Moreover, for this setting, the model proposed in (Hein et al., 2019) yields an AUROC of 0.850 (Table 1 in (Hein et al., 2019)). For this setting, the OE model yields an AUROC of 0.918 for zero-shots (Table 7 in (Hendrycks et al., 2019)).
> > > For normal class SVHN and test set CIFAR-10, FROB with the boundary achieves AUROC 0.958 for zero-shots (Table 14). For this setting, (Lee et al., 2018a) achieves AUROC 0.626 for zero-shots (Table 1 in (Lee et al., 2018a)). (Hein et al., 2019) achieves AUROC 0.938 (Table 1 in (Hein et al., 2019)). The OE model achieves AUROC 0.980 for zero-shots (Table 7 in the Appendix of (Hendrycks et al., 2019)).
> > > When the normal class is SVHN and the test set is *CIFAR-100*, then FROB with the generated boundary samples yields an AUROC of 0.951 for zero-shots (Table 14). For this setting, the model proposed in (Hein et al., 2019) yields an AUROC of 0.935 (Table 1 in (Hein et al., 2019)).
> > >
> > > >The paper uses 80M tiny images dataset, which has been retracted by the original creators.
> > >
> > > The key contribution of our paper is that FROB redesigns and streamlines the use of OoD Datasets to work for few-shot samples, even for zero-shots, using self-supervised learning to model the boundary of the support of the normal class distribution instead of using a large OoD Dataset. Large OoD Datasets are an ad hoc choice of outliers and are used to model the complement of the support of the normal class distribution. The boundary of the support of the normal class data distribution has and needs less samples than the entire full complement of the support of the normal class data distribution. We have fairly compared FROB with several benchmarks, including GOOD (Bitterwolf et al, 2020), ACET, CEDA, the OE model, and others, which use the 80 Million Tiny Images set. Their models are published online. To fairly compare with benchmarks, we have used 80 Million Tiny Images as an OoD Dataset in addition to SVHN and CIFAR-100. The model in (Meinke et al., 2021) uses the 80 Million Tiny Images set.

---

### Decision · Program_Chairs · 2022-01-20

**Decision:**

Reject

**Comment:**

The paper proposes few-shot robust (FROB) model for classification and few-shot OOD detection. While the paper has some interesting contributions, all the reviewers felt that the current version falls short of the ICLR acceptance threshold and the consensus decision was to reject. I encourage the authors to revise the paper based on the reviewers' feedback and resubmit to a different venue.

As Reviewer r838 pointed out, that this paper uses TinyImages dataset which has been since retracted. I appreciate that prior work used TinyImages, but please see "Why it is important to withdraw the dataset" https://groups.csail.mit.edu/vision/TinyImages/ and consider not using the TinyImages dataset for future revisions.